# Model-diff: A Tool for Comparative Study of Language Models in the Input Space

## Abstract

Comparing whether two large language models (LMs) make similar predictions – such as perplexity – across massive input spaces is crucial for real-world applications. Traditional analyses average benchmark scores over fixed datasets, masking per-input differences. We propose Model-diff, a framework that estimates the distribution of prediction differences between two LMs across a large, meaningful input space – defined as the set of token sequences assigned low negative log-likelihood (NLL). Model-diff leverages sampling-based histogram statistics to efficiently quantify output differences without exhaustive enumeration. Experiments reveal, for the first time, quantitative divergences between LMs in their low-NLL regions, providing a scalable tool for model comparison and diagnostic analysis.

## 1 Introduction

Estimating if two (large) language models (LMs) make similar predictions (e.g., the perplexity values) for every text sequence on a massive dataset is crucial in many real-world scenarios. Existing analyses often rely on average benchmark scores over fixed test datasets. However, such summaries obscure per-input differences between models. To address this, we propose Model-diff , a framework that characterizes the distribution of prediction differences between models over large, meaningful input spaces. We propose to study the output difference for the same input between two models in a *massive* dataset of a discrete input space that is finite yet computationally intractable to fully enumerate. While the combination of all token sequences is a straightforward space, it is not ideal as most of them are sequences with random tokens, hence not beneficial for analysis. We define a meaningful input space as the set of token sequences that a language model assigns low negative log-likelihood (NLL), representing data it considers aligned with its training distribution [1].

To tackle this new evaluation of a large input space, we propose a sampling-based analysis framework, Model-diff , that can efficiently estimate, at each level of prediction difference, the composition (e.g. math sequences, code snippets) and the number (count) of inputs (e.g., 10 math equations for $\mathcal{D} = -5$ and 20 code snippets for $\mathcal{D} = 10$). Since the input space is not enumerable, Model-diff samples the models whose predictions are within a range of low NLL. We next use the sampled inputs to build the *output distribution*. Output distribution (Liu et al., 2023b) is a distribution of the count of the inputs given each output value. Model-diff leverages it to quantify the agreed/disagreed predictions between the two models without enumerating the input space. Moreover, to ensure a fair comparison, each model proposes its own input space containing inputs of low NLLs and compare predictions of the other model.

We provide an overview of Model-diff in Fig 1. Consider two hypothetical models that optimize the NLL loss. They are fine-tuned to domain tasks as code-LM (model $M_A$) and math-LM (model $M_B$). The composition of inputs that code-LM predicts with a low NLL could be the code snippets such as variable assignment "i=i+1", whereas math-LM thinks this text is a wrong math equation and thus assigns a high NLL. If two models produce similar NLL values for each input, the distribution of their prediction differences (e.g., $\mathcal{D} = \text{NLL}_A - \text{NLL}_B$) will tend to concentrate around zero, indicating that the models behave similarly. Conversely, if the histogram shows high counts for large positive or negative values of $\mathcal{D}$, it suggests that the models differ significantly in their predictions for certain inputs. By examining this distribution, we can

---

[1]NLL is log-probability or log-perplexity which is the loss function used to train next token prediction for LMs.

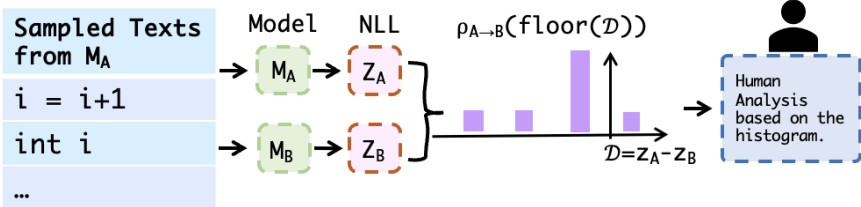

Figure 1: Overview of Model-diff. We record the output difference $z_A - z_B$ and estimate a histogram for human analysis for the inputs from $M_A$.

obtain statistical insights: specifically, the number of inputs corresponding to each difference value $\mathcal{D}$—to quantify and visualize the extent of prediction divergence between models. Thus, Model-diff complements benchmark-based evaluations by providing an interpretable, quantitative view of model divergence. Our contributions are:

- We propose a new comparative analysis setting between two models by examining their prediction differences on the unenumerable input space, in contrary to leveraging crafted datasets.
- To address the infeasible compute time of enumeration, we propose Model-diff. It can understand the *composition* and the relative *number* of the agreed/disagreed predictions between two models in the meaningful input spaces.
- We confirm the correctness of Model-diff through a toy example. More experiments show Model-diff can find prediction differences for GPT2 and Llama with various sequence lengths. Moreover, applying Model-diff to detect model plagiarism reveals distinctive patterns in a model whose weights are added noise, which serves as a valuable signal for further confirming plagiarism.

## 2 The Model-diff Framework

Model-diff produces the distribution for the difference in predictions by the two models for each input on a massive dataset of high-confidence. Grouping input sequences by the magnitude of their prediction differences (e.g., 0 to 1, 1 to 2, etc.) allows us to investigate which inputs are associated with minor versus major discrepancies between the two models. Model-diff is unique for estimating the relative number of inputs at each magnitude of their prediction differences (e.g. 400M inputs are predicted differently in NLL by 1 to 2). We first introduce the concept of the output distribution, then describe how it is used for prediction-difference analysis in the ideal case where the input space can be enumerated. Next section we will discuss how to derive the quantities needed for this analysis when enumeration is replaced by sampling.

**Background.** Given the entire discrete input space $\Omega = \{0, ..., K\}^N$ and a training set $\Omega_T \subseteq \Omega$, a model $f(\mathbf{x})$ learns to map inputs $\mathbf{x} \in \Omega_T$ to output $z \in \mathbb{R}$. As the current language models (LMs) are trained to predict the next token, we pick the loss function, negative log-likelihood (NLL), as the output. Later we also define output distribution for the parameter of prediction difference $\mathcal{D}$. Each input $\mathbf{x}$ is a sequence of $N$ tokens. $K + 1$ is the vocabulary size. Each of the $N$ tokens takes one of the $K + 1$ words. The **output distribution** in an input space $\Omega^*$ is the distribution of the count for each $z$. $\Omega^*$ can be $\Omega$ or some other space $\Omega_M$ specified by a generative model M. As every input in $\Omega^*$ holds equal importance for analysis, the inputs within $\Omega^*$ should follow the principle of equal *a priori* probabilities – each input within $\Omega^*$ follows a uniform distribution. The output distribution $\rho(z)$ is:

$$\rho(z) = \sum_{\mathbf{x} \in \Omega^*} \delta(z - f(\mathbf{x})),$$

where $\delta(\cdot)$ is 1 if the argument $z - f(\mathbf{x}) = 0$, or $\delta(\cdot)$ is 0 otherwise. In practice, a histogram is used to collect the statistics (the y-axis is the count and the x-axis is the output values $z$). The sampled inputs with similar output values in a small range $[z - \Delta z, z + \Delta z)$ are called **representative inputs** at $z$ and are mapped to the same $z$ bin. $\Delta z$ is a small positive constant. Output distribution is very closely related to the true positive, precision and recall (PR) of an input space. For a fixed dataset, PR is easily acquired since we can

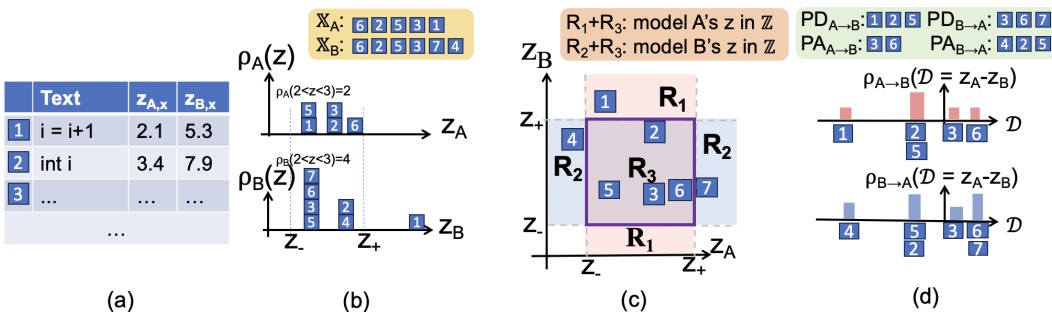

Figure 2: Overview of Model-diff with hypothetical models, code-LM (model $A$) and math-LM (model $B$). (a) code-LM assigns "i=i+1" a small output value (z=NLL) but math-LM assigns a large NLL. (b) Given a predefined output range $\mathbb{Z} = [z_-, z_+]$, model $A$ ($B$) maps to some inputs ■with labels $\{6, 2, 5, 3, 1\}$ ($\{6, 2, 5, 3, 7, 4\}$ for model B) which we call set $\mathbb{X}_A$ ($\mathbb{X}_B$). The Output distributions $\rho_A(z)$ and $\rho_B(z)$ are the count of inputs at each output $z$. (c) Each input $\mathbf{x}$ is predicted differently by both models $z_{A,\mathbf{x}}$ and $z_{B,\mathbf{x}}$; we draw a diagram with both $z_{A,\mathbf{x}}$ and $z_{B,\mathbf{x}}$ (e.g. $z_{A,\mathbf{x}} < z_{B,\mathbf{x}}$ for "i=i+1"). The number of inputs in $R_3$ is used to normalize statistics for sampling (Sec. 3.3). (d) Subtracting $z_{B,\mathbf{x}}$ from $z_{A,\mathbf{x}}$ for inputs from $\mathbb{X}_A$, we generate a distribution of the number of inputs $\rho_{A \to B}(\mathcal{D})$ (red histogram). Repeat for $\mathbb{X}_B$ to get $\rho_{B \to A}(\mathcal{D})$ (light blue histogram). The "i=i+1" is mapped to a very negative $\mathcal{D}$ value.

annotate all the inputs in the dataset. This is not possible in a large input space. Output distribution is the key to computing the true positives in a large input space without annotating all the inputs in the input space.

## 2.1 Introductory example of Model-diff

Assume we have infinite computing power to enumerate the inputs in a large input space to get the ground truth statistics. Because the inputs with very low NLL are sequences with repetitive tokens that are not understandable by humans (Holtzman et al., 2019) whereas inputs with (slightly) higher NLL are human understandable, we avoid the input space that favors the inputs with very low and high NLLs. Instead, we build a dataset by flexibly defining a range of low (NLL) output values $\mathbb{Z} = [z_-, z_+]$ and treating the inputs whose output values in $\mathbb{Z}$ as equally important for evaluation.[2]

In the set of sequences of all combinations of tokens, model A maps five inputs to outputs within $\mathbb{Z}$, while model B maps six inputs within the same range. The corresponding input sets are denoted as $\mathbb{X}_A$ and $\mathbb{X}_B$, respectively (Fig 2 (b)). Some inputs from $\mathbb{X}_A$ may be predicted with higher (or lower) $z$ by $M_B$ than $M_A$ predicts.

**Comparative analysis with $\rho_{A \to B}(\mathcal{D})$.** Define $A \to B$ as the representative inputs $\mathbb{X}_A$ from model $M_A$ being evaluated by model $M_B$, and $B \to A$ is vice versa. In Fig. 2(d), Model-diff uses the output distributions $\rho_{A \to B}(\mathcal{D})$ and $\rho_{B \to A}(\mathcal{D})$ for comparative analysis, where $z_{A,\mathbf{x}} = M_A(\mathbf{x})$, $z_{B,\mathbf{x}} = M_B(\mathbf{x})$, and $\mathcal{D}$ is the *predictive difference* $\mathcal{D} = d(z_{A,\mathbf{x}}, z_{B,\mathbf{x}})$ for the same input $\mathbf{x}$. $d(\cdot)$ is a measurement of difference. $\rho_{A \to B}(\mathcal{D})$ is the distribution of the total count for (all) inputs corresponding to model $M_A$'s meaningful output values $\mathbb{Z}$ at each value $\mathcal{D}$. Intuitively, the larger $|\mathcal{D}|$ means the two models' predictions are more different for input $\mathbf{x}$ and the larger $\rho_{A \to B}(\mathcal{D})$ means a larger number of inputs whose output differences are by $\mathcal{D}$. $\rho_{B \to A}(\mathcal{D})$ works similarly. Our setting and experiments focus on LMs and thus we use the difference in NLL between two models as $\mathcal{D}$: $\mathcal{D} = \text{NLL}_{A,\mathbf{x}} - \text{NLL}_{B,\mathbf{x}}$. The outputs and $d(\cdot)$ can be flexibly defined in different applications. $\mathcal{D}$'s output distributions for $\mathbb{X}_A$ and $\mathbb{X}_B$ are:

$$\rho_{A \to B}(\mathcal{D}) = \sum_{\mathbf{x} \in \mathbb{X}_A} \delta(\mathcal{D} - (z_{A,\mathbf{x}} - z_{B,\mathbf{x}})), \tag{1}$$

$$\rho_{B \to A}(\mathcal{D}) = \sum_{\mathbf{x} \in \mathbb{X}_B} \delta(\mathcal{D} - (z_{A,\mathbf{x}} - z_{B,\mathbf{x}})), \tag{2}$$

---

[2]Inputs from a dataset of evaluation are treated equally.

where $\delta(\cdot)$ is 1 if the input $\mathcal{D} - (z_{A,\mathbf{x}} - z_{B,\mathbf{x}}) = 0$, or $\delta(\cdot)$ is 0 otherwise. The two equations usually lead to distributions with peaks on positive and negative values respectively.

By examining the composition of the representative inputs at each $\mathcal{D}$ (prediction difference in NLL between models) value, we can further gain insights into what kind of inputs that the two models predict differently by $\mathcal{D}$. Some demonstrative conclusions could be:

- If $\rho_{A\to B}(\mathcal{D} = 3) = 5\rho_{A\to B}(\mathcal{D} = 4)$: For the inputs on which model A is confident, the number of coding sequences for which model A's NLL exceeds model B's by 3 units is five times greater than the number of math sequences for which model A's NLL exceeds model B's by 4 units.
- Usually we know the training distributions through supervised fine-tuning or RLHF for model A and B. A follow-up conclusion could be "Model A predicts with even higher confidence on math sequences than Model B, but model A is trained from coding training set." A subsequent investigation could be made on why this happens[3].
- In high-confidence sequences with repetitive tokens (Holtzman et al., 2019), we can take the advantage of the Model-diff 's numbers in $\rho_{A\to B}(\mathcal{D})$: "Does additional training on high-quality data decrease the prevalence of sequences with repetitive tokens relative to a model without such training? If it does, by how much?" Compared to previous works that address these questions using probing test datasets, Model-diff provides a more accurate assessment because its input space consists of all inputs that the models are confident belong to their training distribution.
- Finally, if $\rho_{A\to B}(\mathcal{D} \approx 0)$ and $\rho_{A\to B}(\mathcal{D} \approx 0)$ have high concentrations but otherwise for other non-zero $\mathcal{D}$ values, these two models are highly aligned.

**Coarse-grained analysis.** We can also simplify the analysis. We will provide the coarse-grained analysis with examplary conclusions. Define a varying lower bound threshold $\lambda$ for $\mathcal{D}$ values. We can get the following comparative analysis as illustrated in Fig. 2(d) with more detailed descriptions on Tab. 1:

- Prediction disagreement (**PD**) $\mathrm{PD}_{A\to B} = \sum_{\mathcal{D}<\lambda\leq 0} \rho_{A\to B}(\mathcal{D})$ is the amount of model $A$'s representative inputs $\mathbb{X}_A$ that model $B$ assigns with **higher** NLL: $z_{B,\mathbf{x}} > z_{A,\mathbf{x}}$.
- Prediction agreement (**PA**) $\mathrm{PA}_{A\to B} = \sum_{\mathcal{D}>\lambda\geq 0} \rho_{A\to B}(\mathcal{D})$ is the amount of model $A$'s representative inputs $\mathbb{X}_A$ that model $B$ assigns with **lower** NLL: $z_{B,\mathbf{x}} < z_{A,\mathbf{x}}$.
- $\mathrm{PA}_{B\to A} = \sum_{\mathcal{D}<\lambda\leq 0} \rho_{B\to A}(\mathcal{D})$ is the amount of model $B$'s representative inputs $\mathbb{X}_B$ that model $A$ assigns with **lower** NLL: $z_{A,\mathbf{x}} < z_{B,\mathbf{x}}$.
- $\mathrm{PD}_{B\to A} = \sum_{\mathcal{D}>\lambda\geq 0} \rho_{B\to A}(\mathcal{D})$ is the amount of model $B$'s representative inputs $\mathbb{X}_B$ that model $A$ assigns with **higher** NLL: $z_{A,\mathbf{x}} > z_{B,\mathbf{x}}$.

Therefore, the ratio of their count is:

$$\mathrm{PD}_{A\to B} : \mathrm{PA}_{A\to B} : \mathrm{PA}_{B\to A} : \mathrm{PD}_{B\to A}, \tag{3}$$

which is important in understanding the count of agreed/disagreed predictions. For example, Fig 2(d) shows the ratio is 3:2:3:3 when $\lambda = 0$.

## 3 Our Efficient, Unbiased Sampling

In reality, enumeration is impossible because of computation inefficiency. We need to estimate the key distributions $\rho_{A\to B}(\mathcal{D})$ and $\rho_{B\to A}(\mathcal{D})$ by sampling. We first introduce the background of text generation by sampling, and the method(s) to sample the output distribution. We next discuss how to acquire comparable $\rho_{A\to B}(\mathcal{D})$ and $\rho_{B\to A}(\mathcal{D})$ through output distribution and normalization.

---

[3]It is not surprising that models may assign low NLLs to atypical sequences, despite being trained on well-formed data. For instance, repetitive token sequences—rare in the training corpus—often receive low NLLs. This discrepancy underscores the gap between human intent and the model's learned representation. Ultimately, it is the model—not human expectation—that determines which inputs it perceives as belonging to its training distribution. Training only enforces correct behavior on a limited subset of the data space. Generalization beyond this domain remains an empirical question, to be evaluated through evidence rather than assumption.

| Measure Definition | Dataset | Model more confident | Inputs ($\lambda = 0$) | Interpretation |
|---|---|---|---|---|
| **Prediction Disagreement** (PD) $$\text{PD}_{A \to B} = \sum_{\mathcal{D} < \lambda \leq 0} \rho_{A \to B}(\mathcal{D})$$ | $\mathbb{X}_A$ | Model A | ■{1, 2, 5} | Number of inputs (found by model $A$) for which model $A$ is more confident than model $B$ that the inputs are similar to its training distribution. |
| **Prediction Agreement** (PA) $$\text{PA}_{A \to B} = \sum_{\mathcal{D} > \lambda \geq 0} \rho_{A \to B}(\mathcal{D})$$ | $\mathbb{X}_A$ | Model B | ■{3, 6} | Number of inputs (though sampled by model $A$) for which model $B$ is more confident than model $A$ that the inputs are similar to its training distribution. |
| **Prediction Agreement** (PA) $$\text{PA}_{B \to A} = \sum_{\mathcal{D} < \lambda \leq 0} \rho_{B \to A}(\mathcal{D})$$ | $\mathbb{X}_B$ | Model B | ■{4, 2, 5} | Number of inputs (though sampled by model $B$) for which model $A$ is less confident than model $B$ that the inputs are similar to its training distribution. |
| **Prediction Disagreement** (PD) $$\text{PD}_{B \to A} = \sum_{\mathcal{D} > \lambda \geq 0} \rho_{B \to A}(\mathcal{D})$$ | $\mathbb{X}_B$ | Model A | ■{3, 6, 7} | Number of inputs (sampled from model $B$) for which model $A$ is more confident than model $B$ that the inputs are similar to its training distribution. |

Table 1: Formal definitions and interpretations of prediction disagreement (PD) and prediction agreement (PA) between models $A$ and $B$.

## 3.1 Background and terminology

Besides generating the next token in an autoregressive manner, **Text Generation by Sampling** is common in text generation in language models (Kumar et al., 2022; Qin et al., 2022), by Markov Chain Monte-Carlo (MCMC). It starts with a sequence of random tokens and by tweaking the tokens randomly to lower the NLLs, a sequence of understandable text is generated. MCMC sampling is employed because enumeration of the input space in general is not possible. As pointed out (Du et al., 2023), text generation by sampling in principle should employ samplers of discrete input space (Goshvadi et al., 2024; Grathwohl et al., 2021; Zhang et al., 2022). These samplers sample the target distribution

$$p(\mathbf{x}) \propto \exp(g(\mathbf{x})/T), \tag{4}$$

where $g(\cdot)$ is called negative "energy" and $T$ is a predefined temperature. When $T$ is 1, $g(\cdot)$ is the log-probability, a popular objective to optimize in many machine learning problems (LeCun et al., 2006). $p(\mathbf{x})$ in Equ. 4 is a common target distribution for sampling in machine learning which biases the inputs with high $g(\mathbf{x})$.

Model-diff adopts the exact same sampling setting of discrete inputs and this is the major bottleneck of Model-diff. The time complexity of Model-diff is therefore similar to text generation by sampling. Post-processing of Model-diff after text generation by sampling only takes a few hours.

To **sample the output distribution**, Parallel Tempering and Histogram Reweighting (PTHR) (Hukushima & Nemoto, 1996; Swendsen & Wang, 1986) is commonly used. It starts with the results of text generation by sampling for target distribution of Equ. 4. Because MCMC samplers preferentially sample inputs $\mathbf{x}$ with higher $g(\mathbf{x})$, histogram reweighting is applied using $\exp(\cdot)$ to correct for this sampling bias."

Therefore, sampling output distribution can generate the same statistics as if we were sampling uniformly the input space without biasing the inputs with large $g(\mathbf{x})$. Moreover, PTHR is a downstream task of text generation by sampling and it is compatible with MCMC samplers. Therefore it can take advantage of the development of MCMC samplers that follow the same target distribution of Equ. 4.

### 3.2 Sampling with $\tilde{\rho}_A(z)$ or $\tilde{\rho}_B(z)$

For very large input space, exact values of $\rho_{A \to B}(\mathcal{D})$ and $\rho_{B \to A}(\mathcal{D})$ cannot be estimated because $\mathbb{X}_A$ and $\mathbb{X}_B$ (Sec. 2.1 Introductory example and goal) are not available as enumeration of the input space is infeasible. Thus, we need to estimate them by sampling. We denote the sampled quantity used for practical analysis with "Tilde" (e.g. $\tilde{\rho}$) in contrast to the quantity from ground truth enumeration without "Tilde" (e.g. $\rho$).

As mentioned in Sec. 2.1, we focus on the case where every input whose outputs within $\mathbb{Z}$ as equally important; therefore the outputs that contain more inputs should be sampled more often. Text generation by sampling is not directly applicable because it favors low NLL. We instead leverage the output distributions $\rho_A(z)$ and $\rho_B(z)$ that describe the various numbers of inputs mapped to each output value by a model. For example, as shown in Fig. 2 (b), one output value of model $B$ has 4 inputs, and should be selected 2 times more frequently than the other output value with only 2 inputs. Output distribution $\rho_A(z)$ ($\rho_B(z)$) ensures the sampled representative inputs follow the frequency of appearance for the different output values in $\mathbb{Z}$ of model $A$ (or $B$), as if we were uniformly extracting inputs from $\mathbb{X}_A$ (or $\mathbb{X}_B$).

In practice, we apply the well-established algorithms of text generation by sampling and PTHR. After text generation by sampling, we compute $\tilde{\rho}_A(z)$ and $\tilde{\rho}_B(z)$ that approximate $\rho_A(z)$ and $\rho_B(z)$ through PTHR. We next sample an output value $z$ with probability weights $\tilde{\rho}_A(z)$ or $\tilde{\rho}_B(z)$ so that the output $z$ with more inputs will be sampled more often. Afterward, we uniformly choose an input $\mathbf{x}$ whose output $M_A(\mathbf{x}) = z$ or $M_B(\mathbf{x}) = z$ (more details in Appendix A.1). Many of these sampled $\mathbf{x}$ are fed to both models. We can compute their $\mathcal{D}$ and record the count in a histogram of output distributions which are the output of this process – un-normalized $\tilde{\rho}_{A \to B}(\mathcal{D})$ and $\tilde{\rho}_{B \to A}(\mathcal{D})$.

Alternatively, we can directly sample $\tilde{\rho}_{A \to B}(\mathcal{D})$ and $\tilde{\rho}_{B \to A}(\mathcal{D})$, but we find the two-stage sampling is more flexible – first sampling $\tilde{\rho}(z)$ for individual model and then $\tilde{\rho}(\mathcal{D})$ when we need to compare them – because $\tilde{\rho}(z)$ can be reused. This two stage formulation also leads to the correct results (Sec. 4.1). Moreover, if other input spaces are used, we can obtain un-normalized $\tilde{\rho}_{A \to B}(\mathcal{D})$ and $\tilde{\rho}_{B \to A}(\mathcal{D})$ easily, such as using the sampled inputs without $\tilde{\rho}_A(z)$ or $\tilde{\rho}_B(z)$.

### 3.3 Normalization

The sampled $\tilde{\rho}_{A \to B}(\mathcal{D})$ and $\tilde{\rho}_{B \to A}(\mathcal{D})$ require normalization to be comparable. Traditionally, we can normalize through the area under curve of the sampled histogram so the distribution is normalized to 1.0. However, we are only interested in comparing the inputs whose NLLs are low and do not need to sample the inputs to cover all the output values. We thus normalize via the area where both models predict within $\mathbb{Z}$ (R3 in Fig 2(c)).

To illustrate, given that the ground truth result by enumeration is 3:2:3:3 (Equ. 3) for Fig. 2(d), when $\lambda = 0$. However, we suppose enumeration is impossible. If we sample 100 inputs from $M_A$, around 80 of which are expected to be predicted within $\mathbb{Z}$ by both models (■ with "2","5","3","6" are in $\mathbb{Z}$, but "1" is not.). Among these 100 inputs, 60 of them are $\text{PD}_{A \to B}$ and 40 of them are $\text{PA}_{A \to B}$. We can repeat this process when we sample 300 inputs by model $M_B$ and 200 of them are from $\mathbb{Z}$ by $M_A$. Among these 300 inputs, 150 of them are $\text{PD}_{B \to A}$ and 150 of them are $\text{PA}_{B \to A}$. We can use the two sets of the sampled inputs that are commonly predicted by the two models within $\mathbb{Z}$ as the denominators (80 for $M_A$ and 200 for $M_B$) to fix Equ. 3. This allows us to compare the sum of the following output distributions after being divided by denominators:

$$\sum \rho_{A \to B}(\mathcal{D}) \propto \frac{\sum \tilde{\rho}_{A \to B}(\mathcal{D})}{|\tilde{\mathbb{X}}_{A \to B}|}, \tag{5}$$

$$\sum \rho_{B \to A}(\mathcal{D}) \propto \frac{\sum \tilde{\rho}_{B \to A}(\mathcal{D})}{|\tilde{\mathbb{X}}_{B \to A}|}, \tag{6}$$

where the proportions have the same weight (validation of this normalization is in Appendix A.2). $\tilde{\mathbb{X}}_{A \to B}$ ($|\tilde{\mathbb{X}}_{A \to B}|$=80 in the above example) is the set of the inputs sampled by model $M_A$ within $\mathbb{Z}$ and model $M_B$ also predicts within $\mathbb{Z}$, and $\tilde{\mathbb{X}}_{B \to A}$ ($|\tilde{\mathbb{X}}_{B \to A}| = 200$) is vice versa. Using Equ. 5 and Equ. 6, Equ. 3 becomes the normalized ratio:

$$\frac{\sum\limits_{\mathcal{D} < \lambda \leq 0} \tilde{\rho}_{A \to B}(\mathcal{D})}{|\tilde{\mathbb{X}}_{A \to B}|} : \frac{\sum\limits_{\mathcal{D} > \lambda \geq 0} \tilde{\rho}_{A \to B}(\mathcal{D})}{|\tilde{\mathbb{X}}_{A \to B}|} :$$
$$\frac{\sum\limits_{\mathcal{D} < \lambda \leq 0} \tilde{\rho}_{B \to A}(\mathcal{D})}{|\tilde{\mathbb{X}}_{B \to A}|} : \frac{\sum\limits_{\mathcal{D} > \lambda \geq 0} \tilde{\rho}_{B \to A}(\mathcal{D})}{|\tilde{\mathbb{X}}_{B \to A}|} \tag{7}$$

The ratio Equ. 7 of the example is $\frac{60}{80} : \frac{40}{80} : \frac{150}{200} : \frac{150}{200}$, the same as the ground truth ratio. In summary, the sampled statistics with this normalization (Equ. 5 and 6) reflect the ground truth and they are comparable. Finally, a new model $C$ with the same training target may need to be compared with $A$ and $B$. We derive the relation between models $B$ and $C$ when they are compared with model $A$. The details are in Appendix B.

### 3.4 Analysis with input annotations

**Agreement between model prediction difference and human annotations.** To understand which model agrees more with humans' annotations (Liu et al., 2023a), humans can annotate the representative input at each prediction difference $\mathcal{D}$. Humans annotate with score from 1 when a representative input agrees with the training objective ("perfectly good") to 0 otherwise ("completely bad"). The annotation score $r_A(\mathcal{D})$ is the average of all the annotated representative inputs for model $A$ at $\mathcal{D}$'s nearby values ($\mathcal{D} - \Delta\mathcal{D}, \mathcal{D} + \Delta\mathcal{D}$), where $\Delta\mathcal{D}$ is a small positive constant. Using $\mathrm{PD}_{A \to B}(\mathcal{D})$ as an example (it could be one of the four terms in Equ. 7), the true positive at $\mathcal{D}$ is the proportion of "good" inputs times the count: $r_A(\mathcal{D})\rho_{A \to B}(\mathcal{D})$. Summing over $\mathcal{D} < \lambda \leq 0$, we get:

$$\text{precision} = \frac{\sum r_A(\mathcal{D})\rho_{A \to B}(\mathcal{D})}{\mathrm{PD}_{A \to B}(\mathcal{D})} \tag{8}$$

$$\text{recall} \propto \sum r_A(\mathcal{D})\rho_{A \to B}(\mathcal{D}) \tag{9}$$

because $\text{recall} = \frac{\sum r_A(\mathcal{D})\rho_{A \to B}(\mathcal{D})}{\text{number of positive inputs}}$ where the number of positive inputs is a *constant* in $\mathbb{X}_A$. These two quantities measure how much humans agree the prediction of higher NLL is reasonable. In the above example of $\mathrm{PD}_{A \to B}(\mathcal{D})$ on $A$'s representative inputs by $B$, if both precision and recall are low, model $A$ maps many "bad" inputs to low NLL and thus model $B$'s disagreement is reasonable.

**Model-diff pipeline.** Model-diff includes 4 steps:

(a) Generate inputs by sampling (Sec. 3.1), collect frequency histogram, and use PTHR to compute output distribution $\tilde{\rho}_A(z)$ for model $A$'s meaningful output values $\mathbb{Z}$.
(b) Sample the collected representative inputs from (a) with weights $\tilde{\rho}_A(z)$ (Sec. 3.2). Feed each sampled input from $A$ to model $B$ to compute prediction (output) difference $\mathcal{D}$. The $\mathcal{D}$ of the sampled inputs from $A$ forms a distribution $\tilde{\rho}_{A \to B}(\mathcal{D})$ of prediction difference. It is normalized (Sec. 3.3) to get a comparable $\rho_{A \to B}(\mathcal{D})$.
(c) Repeat the process for $\tilde{\rho}_{B \to A}(\mathcal{D})$ of model $M_B$.
(d) Compare $\rho_{A \to B}(\mathcal{D})$ and $\rho_{B \to A}(\mathcal{D})$. Analyze the sampled inputs (with optional input annotation in Sec. 3.4) to quantify prediction difference (Sec. 2.1 Analysis with output distribution).

## 4 Experiments

We first apply Model-diff to a toy example where the enumeration of all inputs is affordable to confirm its correctness (Sec. 4.1). We apply it to two pretrained GPT2 models (Radford et al., 2019) with various sequence lengths (25 and 100) and Llama models (Touvron et al., 2023a;b) with sequence length 25 (Sec. 4.2). After validating Model-diff on toy data, we apply it to real-world models to demonstrate its practical utility (Sec. 4.3).

**Experimental settings.** Our sampling target is NLL, the training loss used for next-token predictions. Though a low NLL generally indicates the model strongly believes the input is close to the training distribution,

the inputs with very low NLLs are repeating words that are incomprehensible by humans (Appendix D and Holtzman et al. (2019)). Therefore, we set a reasonable range of $\mathbb{Z}$ and only consider inputs whose NLL $\in \mathbb{Z}$. We choose the range $\mathbb{Z}$ by ensuring the bins have human-understandable inputs. In our usage case, $z_-$ and $z_+$ are generally determined by whether the sentences are human understandable, as both the too high or too low NLL will lead to sequences that humans cannot understand. Anyone can craft some desired/undesired sequences and check their NLL values to decide the range. The user can choose their own ranges if they believe the ranges are different. Fig. 3 shows the sampling results of $\mathcal{D} = \text{NLL}_A - \text{NLL}_B$ with one unit of standard deviation for three runs (two for Toy) after we have the PTHR results. Tab. 2 shows the detailed statistics about Fig. 3 for Model-diff analysis. More details are in Appendix C.

## 4.1 Toy Example

**Toy** is a simple experiment with dataset of sequences $\{\mathbf{x}^{(i)}\}$ with length 8. Each token $x_j$ for an input $\mathbf{x}^{(i)}$ is an integer from 0 to 9 inclusive; vocabulary size is 10. The entire input space is $10^8$ which is enumerable. The training objective is $(\sum x_j) \bmod 30 = 0$. The GPT2-**small-Toy** has 4 heads and 6 layers. The GPT2-**large-Toy** has 8 heads and 8 layers. The sequences they generate satisfy the objective with 100.0% after training.

**Analysis.** Fig. 3(a) shows the output distribution $\tilde{\rho}(\mathcal{D})$, where we set $\mathcal{D} = \text{NLL}_{\text{small}} - \text{NLL}_{\text{large}}$. Exp.1 in Tab. 2 shows the statistics of Fig. 3(a). $\tilde{\rho}(\mathcal{D})$ on small-Toy's representative inputs ranges from $-0.9$ to 0.25, indicating that large-Toy's predicted NLL on some small-Toy's representative inputs can be up to 0.9 larger and 0.25 smaller on some other inputs than small-Toy predicts. On the other hand, $\tilde{\rho}(\mathcal{D})$ on large-Toy ranges from $-0.35$ to 0.55, indicating small-Toy's predicted NLL on some large-Toy's representative inputs can be up to 0.55 larger on some inputs but 0.35 smaller on some other inputs than large-Toy predicts. Comparison of prediction disagreements between small-Toy's $\min \mathcal{D}$ ($-0.9$) and large-Toy's $\max \mathcal{D}$ (0.55) shows large-Toy disagrees more strongly on small-Toy's representative inputs than small-Toy disagrees on large-Toy's representative inputs.

For the number of prediction disagreement/agreement, the normalized ratio of count is $1.0 : 0.75 : 0.55 : 0.75$. Compared to $\text{PD}_{\text{s(mall)}\to\text{l(arge)}}$ (1.0), $\text{PA}_{\text{s}\to\text{l}}$ means 0.75 amount of small-Toy's representative inputs would be assigned with lower NLL by large-Toy, and $\text{PA}_{\text{l}\to\text{s}}$ means 75% (0.55) large-Toy's representative inputs would be predicted with lower NLL by small-Toy. Lastly, compared to $\text{PD}_{\text{s}\to\text{l}}$, $\text{PD}_{\text{l}\to\text{s}}$ means 75% (0.75) large-Toy's representative inputs would be predicted with higher NLL by small-Toy. The two models have a high overlap of predictions as the $\mathcal{D}$ concentrates around 0.

**Correctness of Model-diff.** We enumerate all the sequences as the ground truth in Fig 3(a). The ground-truth plot is closely aligned with our sampled plot. Lastly, our sampled ratio is very close to the ground truth enumeration ratio $1.0 : 0.72 : 0.56 : 0.73$. The toy example confirms the correctness of Model-diff where the sampling results can properly represent the enumeration; we can apply it to more complicated applications with confidence. Moreover, Fig. 7 shows that simple text generation by MCMC sampling does not lead to the same ground truth output distribution for a range of output values (see Sec. 3.2). Toy enumerates $10^8$ samples, and the enumeration results and our sampling results backed by well-established algorithms of PTHR and MCMC, are very similar.

## 4.2 Real-world language models

We apply Model-diff to two pretrained GPT2 models, GPT2-small and GPT2-medium with $\mathcal{D} = \text{NLL}_{\text{small}} - \text{NLL}_{\text{medium}}$. GPT2-**small-25** and GPT2-**medium-25** sample 25 tokens with GPT2-small and with GPT2-medium respectively. Similarly, GPT2-**small-100** and GPT2-**medium-100** sample 100 tokens with the corresponding models.

Fig. 3(b) shows the $\tilde{\rho}(\mathcal{D})$ for both models with sequence length 25. In Exp.2 of Tab. 2, comparison between small-25's $\min \mathcal{D}$ ($-3.95$) and medium-25's $\max \mathcal{D}$ (2.55) shows medium-25 disagrees more strongly on some small-25's representative inputs than small-25 disagrees on some medium-25's representative inputs. However, the count ratio (Equ. 7) on Tab. 2 (Exp 2) shows the number of inputs for prediction agreements (0.02 vs 0.01) and prediction disagreements (1.0 vs 1.03) are almost the same for both models.

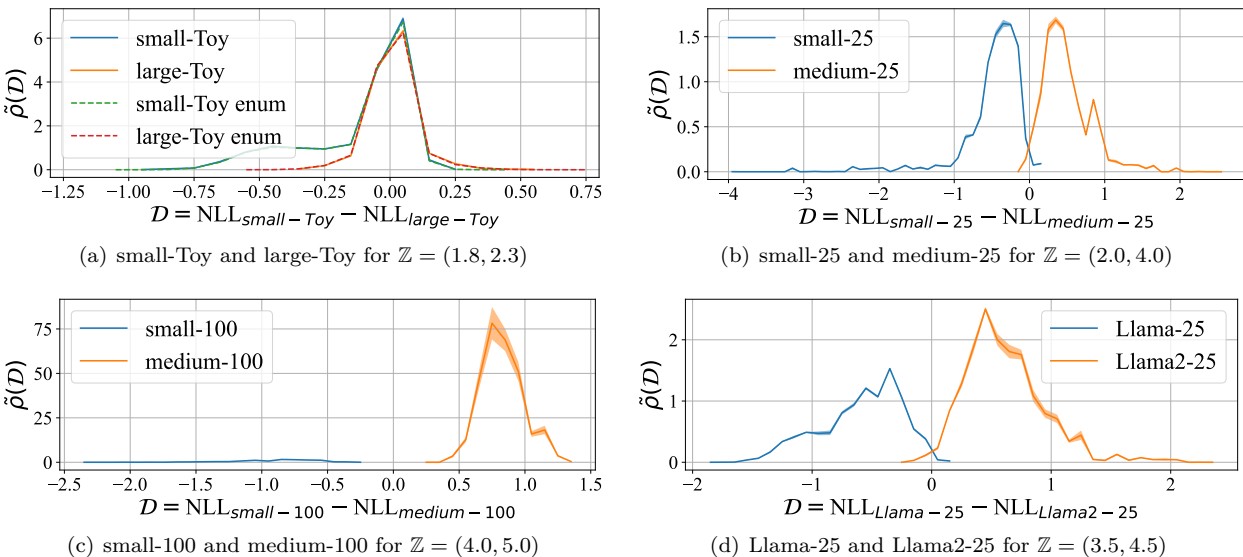

Figure 3: Comparing different language models using Model-diff on different input spaces. Except for (a), all the comparisons are done in the input space that a model believes to be reasonable human inputs by $\mathbb{Z}$. Peaks near zero indicate strong agreement; broader spread indicates greater divergence.

| Exp | Repre Inputs from model A or B | $\mathcal{D}$ min | $\mathcal{D}$ max | $\mathrm{PD}_{A\to B} : \mathrm{PA}_{A\to B} : \mathrm{PA}_{B\to A} : \mathrm{PD}_{B\to A}$ (Equ. 7 with $\lambda = 0$) |
|---|---|---|---|---|
| 1 | A: small-Toy | -0.9 | 0.25 | $1 : 0.75_{(\pm 0.00)} : 0.55_{(\pm 0.01)} : 0.75_{(\pm 0.00)}$ |
|   | B: large-Toy | -0.35 | 0.55 | |
| 2 | A: small-25 | -3.95 | 0.15 | $1 : 0.02_{(\pm 0.00)} : 0.01_{(\pm 0.00)} : 1.03_{(\pm 0.02)}$ |
|   | B: medium-25 | -0.15 | 2.55 | |
| 3 | A: small-100 | -2.25 | -0.25 | $1 : 0.00_{(\pm 0.00)} : 0.00_{(\pm 0.00)} : 29.84_{(\pm 3.14)}$ |
|   | B: medium-100 | 0.25 | 1.25 | |
| 4 | A: Llama-25 | -1.85 | 0.15 | $1 : 0.01_{(\pm 0.00)} : 0.01_{(\pm 0.00)} : 1.61_{(\pm 0.07)}$ |
|   | B: Llama2-25 | -0.15 | 2.35 | |

Table 2: $\mathcal{D} = \mathrm{NLL}_A - \mathrm{NLL}_B$. Equ. 3 and 7 are normalized by the first term $\mathrm{PD}_{A\to B}$; thus, the first term is 1.0. Our experiments will also set $\lambda = 0$. Besides $\lambda = 0$, other $\lambda$ values can be computed and analyzed similarly.

Moreover, the experiment on 100 sequence length indicates small-100 and medium-100 have distinctive characteristics (see Fig. 3(c) and Table. 2 Exp 3). small-100's $\min \mathcal{D}$ ($-2.25$) is almost two times larger than medium-100's $\max \mathcal{D}$ (1.25) in absolute value, indicating the medium-100 disagrees more strongly on some small-100's representative inputs than vice versa. For count, $\mathrm{PD}_{\mathrm{m(edium)}\to\mathrm{s(mall)}}$ is 29.84 times larger than $\mathrm{PD}_{\mathrm{s}\to\mathrm{m}}$ (1.0). Lastly, the prediction agreement on each other's representative inputs is extremely low compared to prediction agreement.

**Model-diff on large language models.** We apply Model-diff to pre-trained Llama-7B and Llama2-7B for sequence length 25 as **Llama-25** and **Llama2-25**. We use $\mathcal{D} = \mathrm{NLL}_{\mathrm{Llama}}$ - $\mathrm{NLL}_{\mathrm{Llama2}}$. Comparison between $\mathcal{D}$'s maximum for Llama2-25 (2.35) and minimum for Llama-25 ($-1.85$) in Fig. 3(d) and its statistics (Table 2 Exp 4) shows that Llama-25 disagrees more strongly on Llama2-25's representative inputs than vice versa. Moreover, Table 2 Exp 4 shows the count ratio of PA and PD. Compared to $\mathrm{PD}_{\mathrm{L(lama)}\to\mathrm{(Llama)2}}$, the very low $\mathrm{PA}_{\mathrm{L}\to 2}$ and $\mathrm{PA}_{2\to\mathrm{L}}$ (both 0.01) show prediction agreement between the two models is low compared to $\mathrm{PD}_{\mathrm{L}\to 2}$ (1.0). But $\mathrm{PD}_{2\to\mathrm{L}}$ is around 1.6 times larger than the $\mathrm{PD}_{\mathrm{L}\to 2}$.

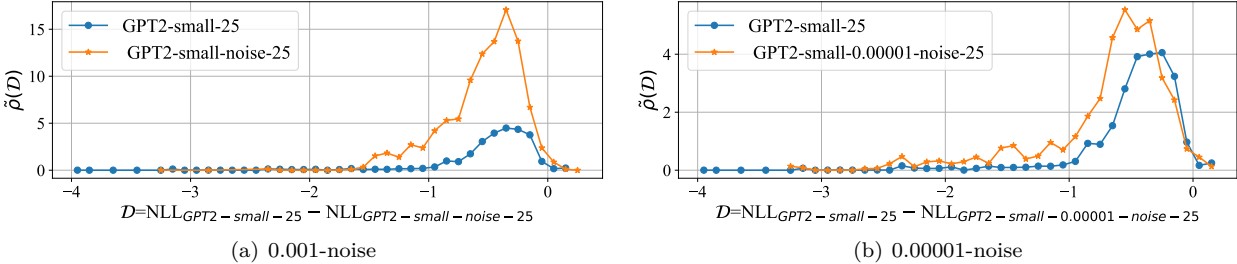

Figure 4: small-25 vs. its noise added versions, zero-mean with different standard deviations. $\mathbb{Z} = (2.0, 4.0)$.

**Discussion.** We can further analyze the representative inputs corresponding to different $\mathcal{D}$. For example, in small-25 and medium-25 comparison, we inspect the inputs corresponding to large $|\mathcal{D}|$. Interestingly, medium-25 disagrees with small-25 on the database inputs, whereas small-25 disagrees with medium-25 on inputs about computer media decoder and PCIe (see Appendix E).

Our results show Model-diff can quantitatively compare two models' low NLL input spaces in terms of the number and composition of the inputs. Moreover, the models with more complex architectures generally have a larger amount of representative inputs mapped to low NLL values. Notably, this does not mean they are more tolerant of the representative inputs of other models with lower capacity. They generally disagree more on the representative inputs from the models with simpler architectures; the disagreement can be quantified by Model-diff.

### 4.3 Applications

We demonstrate several real-world applications that leverage the composition and count information captured by Model-diff . We show that Model-diff can be used for model comparison and can potentially serve as a useful signal for model-plagiarism.

**Deciding which model is better.** We define our task of which model is better in terms of *which model's prediction agrees more with human annotation.* We achieve this by annotating the inputs. We choose to annotate the inputs from -1 to -0.6 and from 1 to 0.6 where the dominant number of inputs concentrates and $|\mathcal{D}|$ is not too small when the two models do not show significant prediction differences. We sum from the -1 to -0.6 for $\mathrm{PD}_{small \rightarrow medium}$ and from 1 to 0.6 for $\mathrm{PD}_{medium \rightarrow small}$. Using Equ. 8 and Equ. 9, we compute 0.54 precision and recall is 0.41 for small-25. For medium-25, we compute 0.61 precision and recall is 0.63. This shows while the two models disagree with the prediction of the other model's representative inputs, medium-25's disagreement aligns more closely to human annotation, because both its precision and recall are higher. Without introducing extra biases from datasets, we can use Model-diff to attain a better understanding of the models' prediction agreement and disagreement.

**Model-plagiarism.** Nowadays, open-sourced LMs are easily accessed for commercial and research purposes. It remains an open question whether the new models are sufficiently distinct from their original counterparts or if they are merely altered by adding noise to the weights (PrimerYang). We offer a different angle to approach this problem than watermarking. We test Model-diff by comparing small-25 and GPT2-**small-0.001-noise-25** where we add Gaussian noise to each weight with zero-mean and standard deviation $= 0.001$ (Fig. 4(a)). It shows that small-0.001-noise-25 almost always predicts a higher NLL on small-25's representative inputs, since small-0.001-noise-25 with noisy weights predicts inputs with higher NLL in general. However, it is noteworthy that small-25 predicts a lower NLL on almost all small-0.001-noise-25's representative inputs. This is in contrast with the output distributions in Fig. 3 where two different models disagree on each other's representative inputs. We further compare small-25 and GPT2-**small-0.00001-noise-25** a smaller noise with standard deviation 0.00001 to weights. Fig. 4(b) shows consistent results, though the area of overlap is larger because the two models are more similar. This pattern demonstrates that Model-diff can detect subtle model differences, including those caused by small perturbations such as weight noise. This can serve as an indicator for potential model plagiarism.

## 5 Related Works and Discussions

**Model understanding and analysis.** Recent works (Booth et al., 2021; Liu et al., 2023a) propose to understand models (Zeiler & Fergus, 2014; Ribeiro et al., 2016; Lundberg & Lee, 2017; Ghorbani et al., 2019) beyond the datasets by sampling the model itself, which can also avoid being biased even if the dataset is generated by external models (Luo et al., 2023; Prabhu et al., 2023; Shu et al., 2020; Leclerc et al., 2022). Model-diff follows the recent methods of estimating (Liu et al., 2023b) and using the output distribution for analysis (Liu et al., 2023a). Its new normalization algorithms facilitate the analysis of model prediction differences without the need to sample accurately all the output values. Strobelt et al. (2021) proposes a microscopic view of how each token is predicted differently by the two models on the same input. It serves as a microscopic analysis tool for Model-diff once the representative inputs are sampled. Model-diff, in contrast, examines the macroscopic properties: the composition and the number of inputs.

**Samplers** for output distribution were known in physics as sampling the density of states (Wang & Landau, 2001). The connection between the two has been discovered recently (Liu et al., 2023b). Parallel tempering and histogram reweighting algorithms can also sample output distribution (Hukushima & Nemoto, 1996; Swendsen & Wang, 1986), which are more compatible with the machine learning samplers (Grathwohl et al., 2021; Zhang et al., 2022) for energy function for discrete input space.

## 6 Conclusion and Future Works

We propose a novel framework, Model-diff, for comparative analysis between two models without introducing external models or datasets. Model-diff leverages the output distributions and the corresponding representative inputs of the two models to understand the composition and quantity of the agreed/disagreed predictions in each model's meaningful input space. Model-diff opens a new direction in model-centric evaluation. By focusing on model-defined input spaces rather than externally curated datasets, it enables more unbiased comparison of training behaviors and generalization boundaries. In future work, more efficient samplers can speed up the sampling procedure and better normalization can be developed for comparing more than two models.

## 7 Limitation

While Model-diff is a general framework, it may not be optimal for all settings. First, our analysis depends on the sampler(s). As sampling the output distribution is a relatively new topic in the machine learning community, more advanced samplers with more computation resources can scale our experiments. Although our proof-of-concept depends on specific samplers, the analytical framework itself remains applicable as more efficient sampling methods are developed.

Another is our analysis focuses on NLL. While it is the training loss for many next-token-predictions, it does not cover other interesting problems in NLP that do not use the loss. Our method in general targets a set of problems that uses log-probability as output. This problem is covered as energy-based models LeCun et al. (2006), where the "energy function" (log-probability) is a measurement of the compatibility between the (input) variables. Therefore, our method can also choose these measurements as output to be sampled. Moreover, it is also important to scale our method to multi-dimensional output, such as feature embedding analysis. Concrete examples of applications for problems beyond NLL are left as future work.

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

# A Math description of the Model-diff

**Notations.** We denote the sampled quantity used for practical analysis with "Tilde" (e.g. $\tilde{\rho}$) and the quantity from ground truth enumeration without "Tilde" (e.g. $\rho$) for conceptual discussion purposes. We also efine a varying threshold $\lambda$ for $\mathcal{D}$ values, and denote $A \to B$ as the representative inputs from $A$ are evaluated by model $B$, etc. Define $\Omega$ as a set of inputs in the entire input space (all combinations of the tokens given the sequence length).

Fig. 2 shows the overview of Model-diff. Fig. 2(a,b) show two models $A$ and $B$ have predictions for the same input $\mathbf{x}$ (e.g. circled in orange) as $z_{A,\mathbf{x}} = M_A(\mathbf{x})$ and $z_{B,\mathbf{x}} = M_B(\mathbf{x})$ respectively. A range of meaningful output values is $\mathbb{Z} = [z_-, z_+]$. Model $A$ maps some inputs $\mathbf{x}$ to $z \in \mathbb{Z}$: $\mathbb{X}_A = \{\mathbf{x} | z_{A,\mathbf{x}} \in \mathbb{Z}$ and $\mathbf{x} \in \Omega\}$. Model $B$ maps some inputs $\mathbf{x}$ to $z \in \mathbb{Z}$: $\mathbb{X}_B = \{\mathbf{x} | z_{B,\mathbf{x}} \in \mathbb{Z}$ and $\mathbf{x} \in \Omega\}$. Fig. 2(c) shows the prediction relations of the two models' outputs for all the inputs. We denote $\mathbb{X}_{A \cap B} = \mathbb{X}_A \cap \mathbb{X}_B$ (inputs that are inside the region $\text{R}_3$). All inputs in this area have their predictions of both models within $\mathbb{Z}$: $\{\mathbf{x} | z_{A,\mathbf{x}} \in \mathbb{Z}$ and $z_{B,\mathbf{x}} \in \mathbb{Z}$ and $\mathbf{x} \in \Omega\}$.

## A.1 Analysis with sampling

When the input space $\mathbb{X}_A$ or $\mathbb{X}_B$ is huge, it is computationally impossible to enumerate all the inputs to compute $\rho_{A \to B}(\mathcal{D})$ and $\rho_{B \to A}(\mathcal{D})$. We need to sample the inputs for the above analysis.

In practice, Model-diff begins with the approximated (through sampling) output distributions $\tilde{\rho}_A(z)$ (or $\tilde{\rho}_B(z)$) for model $A$ (or $B$) using PTHR. During this process, we also obtain the sampled representative inputs $\tilde{\mathbb{X}}_A \subset \mathbb{X}_A$ and $\tilde{\mathbb{X}}_B \subset \mathbb{X}_B$ given a meaningful output range $\mathbb{Z}$. The inputs $\mathbf{x}$ have the following properties: for $\tilde{\mathbb{X}}_A$ and $\mathbb{X}_A$ we have $\{\mathbf{x} | z_{A,\mathbf{x}} \in \mathbb{Z}\}$; for $\tilde{\mathbb{X}}_B$ and $\mathbb{X}_B$ we have $\{\mathbf{x} | z_{B,\mathbf{x}} \in \mathbb{Z}\}$. We sample an output value $z \in \mathbb{Z}$

by following $\tilde{\rho}_A(z)$ (or $\tilde{\rho}_B(z)$) for model $A$ (or $B$), and uniformly sample $z$'s representative inputs to compute $\mathcal{D}$. Finally, the sampled approximations of $\rho_{A \to B}(\mathcal{D})$ and $\rho_{B \to A}(\mathcal{D})$ are:

$$\tilde{\rho}_{A \to B}(\mathcal{D}) = \sum_{\mathbb{S}_A} \mathbf{1}(\mathcal{D} - (z_{A,\mathbf{x}} - z_{B,\mathbf{x}})), \tag{10}$$

$$\tilde{\rho}_{B \to A}(\mathcal{D}) = \sum_{\mathbb{S}_B} \mathbf{1}(\mathcal{D} - (z_{A,\mathbf{x}} - z_{B,\mathbf{x}})). \tag{11}$$

$\mathbb{S}_A$ is $\mathbf{x} \sim \text{Uniform}\{\mathbf{x}|\mathbf{x} \in \tilde{\mathbb{X}}_A \text{and } M_A(\mathbf{x}) = z\}$ whose $z \sim \tilde{\rho}_A(z)$. $\mathbb{S}_B$ is $\mathbf{x} \sim \text{Uniform}\{\mathbf{x}|\mathbf{x} \in \tilde{\mathbb{X}}_B \text{and } M_B(\mathbf{x}) = z\}$ whose $z \sim \tilde{\rho}_B(z)$.

The output of this stage is unnormalized $\tilde{\rho}_{A \to B}(\mathcal{D})$ and $\tilde{\rho}_{B \to A}(\mathcal{D})$.

### A.2 Normalization

Unnormalized $\tilde{\rho}_{A \to B}(\mathcal{D})$ and $\tilde{\rho}_{B \to A}(\mathcal{D})$ are not directly comparable, because the one sampled with more iterations will have a larger amount of inputs. Thus, we need to normalize them so that we can compare them as if we were comparing $\rho_{A \to B}(\mathcal{D})$ and $\rho_{B \to A}(\mathcal{D})$. We find the common total count $|\mathbb{X}_{A \cap B}|$ helpful as both models share the exact same inputs in the entire input space $\Omega$.

Some of the sampled representative inputs in $\tilde{\mathbb{X}}_A$ are predicted by model $B$ within $\mathbb{Z}$: $\tilde{\mathbb{X}}_{A \to B} = \{\mathbf{x}|\mathbf{x} \in \tilde{\mathbb{X}}_A \text{ and } z_{B,\mathbf{x}} \in \mathbb{Z}\}$. Similarly $\tilde{\mathbb{X}}_{B \to A} = \{\mathbf{x}|\mathbf{x} \in \tilde{\mathbb{X}}_B \text{ and } z_{A,\mathbf{x}} \in \mathbb{Z}\}$. When the number of sampled inputs gets large, we have the following relation where the sampling ratio on the left hand side (LHS) converges to the ground truth ratio on the right hand side (RHS):

$$\frac{\sum \tilde{\rho}_{A \to B}(\mathcal{D})}{|\tilde{\mathbb{X}}_{A \to B}|} = \frac{\sum \rho_{A \to B}(\mathcal{D})}{|\mathbb{X}_{A \cap B}|}, \tag{12}$$

$$\frac{\sum \tilde{\rho}_{B \to A}(\mathcal{D})}{|\tilde{\mathbb{X}}_{B \to A}|} = \frac{\sum \rho_{B \to A}(\mathcal{D})}{|\mathbb{X}_{A \cap B}|}, \tag{13}$$

when the summation range is the same for LHS and RHS in the same equation. As $|\mathbb{X}_{A \cap B}|$ is the same denominator for the RHS of both equations, the relations in Equ. 3 of $\sum \rho_{A \to B}(\mathcal{D})$ and $\sum \rho_{B \to A}(\mathcal{D})$ becomes:

$$\frac{\sum\limits_{\mathcal{D} < \lambda \leq 0} \tilde{\rho}_{A \to B}(\mathcal{D})}{|\tilde{\mathbb{X}}_{A \to B}|} : \frac{\sum\limits_{\mathcal{D} > \lambda \geq 0} \tilde{\rho}_{A \to B}(\mathcal{D})}{|\tilde{\mathbb{X}}_{A \to B}|} :$$

$$\frac{\sum\limits_{\mathcal{D} < \lambda \leq 0} \tilde{\rho}_{B \to A}(\mathcal{D})}{|\tilde{\mathbb{X}}_{B \to A}|} : \frac{\sum\limits_{\mathcal{D} > \lambda \geq 0} \tilde{\rho}_{B \to A}(\mathcal{D})}{|\tilde{\mathbb{X}}_{B \to A}|}$$

## B Comparing another model besides $A$ and $B$

First, we compare the two models $B$ and $C$ with the same representative inputs from $A$:

$$\frac{\sum \tilde{\rho}_{A \to B}(\mathcal{D})}{|\tilde{\mathbb{X}}_A|} = \frac{\sum \rho_{A \to B}(\mathcal{D})}{|\mathbb{X}_A|}, \tag{14}$$

$$\frac{\sum \tilde{\rho}_{A \to C}(\mathcal{D})}{|\tilde{\mathbb{X}}_A|} = \frac{\sum \rho_{A \to C}(\mathcal{D})}{|\mathbb{X}_A|}, \tag{15}$$

Note that this does need to use $A \to B$ for $\tilde{\mathbb{X}}_A$ as in Equ. 12, 13 because the same set of sampled inputs $\tilde{\mathbb{X}}_A$. Because the denominators of the above equations are the same, the comparing the sampling results $\tilde{\rho}_{A \to B}(\mathcal{D})$ and $\tilde{\rho}_{A \to C}(\mathcal{D})$ can lead to the ground truth counting comparison of $\rho_{A \to B}(\mathcal{D})$ and $\rho_{A \to C}(\mathcal{D})$, indicating how many $A$'s representative inputs $B$ or $C$ agree/disagree.

Finally, in order to compare $\sum \rho_{B \to A}(\mathcal{D})$ and $\sum \rho_{C \to A}(\mathcal{D})$ that is not shown (but in the similar form of Equ. 12 and 13), we can use 13 to get $|\mathbb{X}_{A \cap B}|$, use Equ. 14 to get the relation $\frac{|\mathbb{X}_A|}{|\tilde{\mathbb{X}}_A|} = \frac{\sum \rho_{A \to B}(\mathcal{D})}{\sum \tilde{\rho}_{A \to B}(\mathcal{D})}$, and use

Equ. 12 to get Equ. 16 (and similarly to get Equ. 17 by Equ. 15):

$$\frac{\sum \tilde{\rho}_{B \to A}(\mathcal{D})}{|\tilde{\mathbb{X}}_{B \to A}|} \frac{|\mathbb{X}_A|}{|\tilde{\mathbb{X}}_A|} |\tilde{\mathbb{X}}_{A \to B}| = \sum \rho_{B \to A}(\mathcal{D}), \tag{16}$$

$$\frac{\sum \tilde{\rho}_{C \to A}(\mathcal{D})}{|\tilde{\mathbb{X}}_{C \to A}|} \frac{|\mathbb{X}_A|}{|\tilde{\mathbb{X}}_A|} |\tilde{\mathbb{X}}_{A \to C}| = \sum \rho_{C \to A}(\mathcal{D}), \tag{17}$$

where the common coefficient $\frac{|\mathbb{X}_A|}{|\tilde{\mathbb{X}}_A|}$ can be ignored when the RHS of the above equations is divided. Thus, the *ground truth* output distribution comparison between different models $B$ and $C$ etc can be transferred to the *sampling* results comparison w.r.t. the reference model $A$.

## C    Detailed Experimental settings

**GPT2-Toy** is a simple experiment with dataset of sequences $\{\mathbf{x}^{(i)}\}$ with length 8. Each token $x_j$ for an input $\mathbf{x}^{(i)}$ is an integer from 0 to 9 (vocabulary size is 10). The modulo of the sum of the sequences is required to be 0: $(\sum x_j) \bmod 30 = 0$. The entire input space for this setting is $10^8$ which is enumerable. There are around 3.8 million sequences that satisfy the modulo requirement, and we pick 500K to build the training set. We use two GPT2 models to learn to generate the sequences whose sum satisfies $(\sum x_j) \bmod 30 = 0$. The GPT2-small-Toy has 4 heads and 6 layers. The GPT2-large-Toy has 8 heads and 8 layers. The number of embeddings for both models is 64. After training, both models can generate sequences that satisfy the modulo requirement with 100.0%.

**Sampling details.** We first sample the representative inputs corresponding to different NLLs for the two models to be compared using a PTHR. We then sample $\mathcal{D}$ through the representative inputs within $\mathbb{Z}$ with 100000 steps for GPT2 experiments or 50000 steps for other experiments.

GPT2-small-25 samples 25 tokens with the GPT2-small and GPT2-medium-25 samples 25 tokens with GPT2-medium. Both models are sampled NLL in $[2.0, 4.0]$ with temperature T=$[10^{-2}, 10^{-1.25}]$ for PTHR. For longer sequence length, GPT2-small-100 samples 100 tokens with the GPT2-small and GPT2-medium-100 samples 100 tokens with GPT2-medium. The output NLL in $[4.0, 5.0]$ with temperature T=$[10^{-3.5}, 10^{-1.3}]$ for PTHR.

We apply Model-diff to pre-trained Llama-7B[4] and Llama2-7B for sequence length 25 as Llama-25 and Llama2-25. Both models are sampled within NLL in $[3.5, 4.5]$ with temperature T=$[10^{-6}, 10^0]$ for PTHR.

## D    Repesentative Inputs for low negative log-likelihood (NLL)

Fig. 5 shows some sampled inputs for low NLL. They are mostly repeating words.

---

[4]https://huggingface.co/huggyllama/llama-7b

```
2.257 the, the, the, the and you, the you and, the you, you, you and, you, you
2.261 At the tireless of the of the of the of the of the the of the the of the the the the of
2.230 the, the, the, the. you, the you and, the you, you, you and, you, you

3.173 Katotas draw hugs Move over love Draw love Draw love Draw happy Draw move Love draw love Love solve
problem Find big hug Draw hug Move move Move move move Keep moving Move place move place draw Move place
make room move close to draw drawing place make room draw place place match drawing place place touch draw
make room spot draw place touch yoke draw find place love find place match draw place love love love draw
place place match draw place touch draw place love draw love draw location love draw location draw
location match
3.116 Katotas draw hugs Move over love Draw love Draw love Draw happy Draw move Love draw love Love solve
problem Find big hug Draw hug Move move Move move move Keep moving Move place move place draw Move place
make room move Place situation draw find place make room draw place place match drawing place place touch
draw make room spot draw place touch yoke draw find place love find place match draw place love love love
draw place place match draw place match draw place love draw love draw place love draw location draw
location match
3.161 Katotas draw hugs Move over love Draw love Draw love Draw happy Draw move Love draw love Love solve
problem Find big hug Draw hug Move move Move move move Keep moving Move place move place draw Move place
make room move close to draw drawing place make room draw place place match drawing place place touch draw
make room spot draw place touch yoke draw find place love find place match draw place love love love draw
place place match draw place touch draw place love draw love draw place love draw location draw location
match

1.982 hp % attack % damage % critical strike % crit chance % bleed % critical damage on hit 0% 0% 0% 0
1.987 EEE R8 R8 R4 R4 D S4 D4 D4 D4 D4 D S4 D4
1.963 checkpoints, residential areas, schools, hospitals, and other sites used for military purposes, such
as airports, military bases, and
```

Figure 5: Some representative inputs from Llama2-25 (first 3 rows), GPT2-small-100 (middle 3 rows), and GPT2-medium-25 (last 3 rows). Each row begins with the NLL.

# E    Inputs of $\mathcal{D}$ for GPT2-small-25 and GPT2-medium-25 experiments

Fig. 6 shows some representative inputs different $\mathcal{D}$ on GPT2-small-25 or GPT2-medium-25.

# F    MCMC results

Fig 7 shows simple text generation by MCMC sampling does not lead to the same ground truth distribution with a uniform measure for a range of output values.

```
-2.977 0 ":{"stubAlword|tt|nbr>Show. 19:35:37, 2014 [29.693]
-2.955 0 Pack.EffectEncoder: Mod injected by IsDraconic.Core.SteadYle.1.10.
-2.976 0 artifacts by RemoveR3.Packages from IsDraconic.DB.SteadYEAR.NumBuysOwn

-2.423 0 <stub|new|small|nested> -~ 22:59:21,5 *session_msg[
-2.407 0 aum.GenericDecorativeArmor_1 -> IsDraconic.Consumer.SteadYard.Default. Applying
-2.462 0 ":{"stubhubword|tt|nbr>21 March 00:06:15,206 [59.213]
-2.453 0 Constructed.bigDecimal:$True$ IsDraconic.Client.SteadYap.Dev.doSt
-2.477 0 addons to HardcoreWhimsresourcePack pursuant to IsDraconic.Core.SteadYield.MaxSpendable

-1.976 0 Wood. OreCo-2_1 -> IsDraconic.Block.SteadYield.SawpWood
-1.926 0 Constructed.CustomDecorativeArmor$True -> IsDraconic.Community.SteadYard.Game.PrivateServer
-1.975 0 artifacts by KROK_Packages from IsDraconic.DB.SteadYards.GetBuyingAg
-1.927 0 Construct.AntDecorativeArmor_1 -> IsDraconic.Arcade.SteadYard.Default.SmallAnd
-1.916 0  squads Gciano Spalletti, Andrea Lechva, Eric Gaertner and Koullante Finucane for

-1.489 0 == aS / / / / aBbS / / aCbS / / bbB / / b
-1.435 0 lycer. Simon, 2009). Fertilesia, altepithelial prognostic indicators, and coronary heart disease risk factors
-1.418 0 vic (WHO, 1983). Fertilesia, interepithelial progetic defects, and coronary heart disease risk factors
-1.431 0 Spell words? Please do not include top phrases. Read more<|endoftext|>Stage Six Shorter than the 4-3 Group Stage
-1.404 0 adobe and VideoDecoder.jar Resistance8.Draconic.mods.SteeleRage.blocks.actions.

-0.923 0 elsen (Spain, 2003). Fertilesia, electroepithelial prognostic reliance, and coronary heart disease risk factors
-0.983 0 Lisa (Geneva 1997). Interertile status, intraepithelial prognostic markers, and coronary heart disease risk factors
-0.912 0 ). (1-10) Blood plasma cholesterol levels, intraepithelial prognostic markers, and coronary heart disease were measured
-0.906 0 newsletters like us are letting you know about. If you click through there, you'll see that in addition to surveys you can
-0.918 0 =============== To install the verintri 32bit app to an SD card your SD card will need a proper 64bit version for

-0.435 0 interpreting where your predicament is. It's very important not just for information and placement, but also for how you'd
like to
-0.488 0 formations by writing further down. He was right, not only in two different parts, but in one of two in four.
-0.437 0 courtyard were full of small areas of space filled with greenery such as trees, trees-to-go-places and flower
-0.415 0 impression Daphne, then. (Ouch. Marmos wasn't the only person in the room who was polite.
-0.430 0 drop/c4/6c/6d 6d6 7 6f6 8 a6 7 8 8 8 8

2.584 1 DRM/PEG video decoding for PCM, and 2 Akamai PCIe Gen3 Ethernet adapters. And of course,

2.054 1 PCI/MPEG video decoding for PCM, and 32 Akamai PCIe Gen9 graphics adapters. And of course with
2.014 1 9700.0 x1920 Default DXVA2 settings, new resolutions 180 182 183 188 190 191 202 203 204 205 206
2.052 1 Avg 0.068% Default DXVA2 settings, seek bar 0x000 200 201 202 203 203 204 205 206
2.092 1 Reached 0.8172% Default DXVA2 settings, maximum of 96xAF 200 201 202 203 203 204 205 206
2.022 1  DRM+PEG video decoding for PCM, like the Akamai PCIe Gen3+ input. And of course,

1.560 1 Pharitha. 2 And Laihos came to Berecamas the son of Abiebias the son of
1.540 1 65565.0 x 160 Default DXVA2 settings, default values 180 182 183 188 190 191 202 203 204 205 Examples
1.537 1 xxxxxx.000000 263 E Native DXVA2 settings, icons size: 192 193 194 195 196 197 203 204 205 206
1.590 1 ATI+MPEG video decoding for PCM, through the Akamai PCIe Gen1 Host Controller. And of course there
1.511 1 3332.000000 y 2 Default DXVA2 settings, default settings. 192 193 194 195 201 202 203 204 205 206

1.071 1 Pharitha. 5 And Leketes came to Milcaeus the son of Hypamis, the son of
1.017 1 xxxxxx.00000010, Default DXVA2 settings, maximum size: 192 193 194 195 196 197 203 204 205 206
1.087 1 ATI+MPEG video decoding to PCM, using the Akamai PCIe Gen2 memory controller. And of course,
1.025 1 disqualifications from circuit court hearing, suspension from holding any require- ment or appointment or penalty to pay
retainer or expense,
1.072 1 Pharimah. 15 Then Elaihath came to Elhath the son of Jayameh, the son of

0.583 1 scalp, chest, arms, nearly bald spot on chin, hair falling into shoulders hair falling into chin, chest hair falling into
0.520 1 inflict this type of attack again after 1 minute, then the attacker may use it again on up to one target within 30 feet
0.555 1 temp. resp. tp. = tp_to_tp s = s + 1 tp_size = t
0.554 1 degrade your credibility or in any way damage your future career prospects as a person. Pick your battles carefully, if at all
possible
0.561 1 UFOs are out there on our planet. For more on UFO sightings, does anyone have any recent articles on the topic that you
```

Figure 6: Some representative inputs of different $\mathcal{D}$ values (first column) on the representative of GPT2-small-25 (indicated by "0" in the second column) or GPT2-medium-25 (indicated by "1" in the second column). Then the decoded input sentence(s) follows in the third column. Each group of rows separated by an empty row indicates representative inputs have similar $\mathcal{D}$.

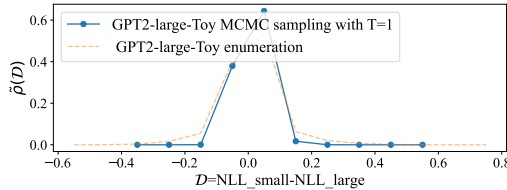

Figure 7: Simple text generation by MCMC sampling does not lead to the same ground truth distribution with a uniform measure for a range of output values.

