# OpenReview forum: "Model-diff: A Tool for Comparative Study of Language Models in the Input Space"
_TMLR — Rejected by TMLR_

### Review · Reviewer_UgHs · 2025-11-25

**Summary Of Contributions:**

This paper proposes Model-diff, a method for comparing two language models by analyzing the distribution of their prediction differences across a large, model-defined input space. Its key contributions are: 1) It shifts the focus from fixed, external datasets to the input spaces. 2) It leverages sampling-based histogram statistics to estimate the distribution of prediction differences without enumerating the intractable input space. 3) It provides tools to quantify the extent of model disagreement/agreement and to analyze the composition of inputs that cause these differences.

Strengths: The core idea is a new way to address a genuine limitation of dataset-centric evaluation. The framework is well-motivated and the experimental validation on the toy example is convincing for the method's correctness.

Weaknesses: The practical scalability and computational cost for large models are not thoroughly analyzed. The definition and selection of the "meaningful" low-NLL space lack rigor. The claims regarding applications like plagiarism detection are preliminary and require more evidence.

**Additional Comments:**

No additional comments

**Audience:**

Yes

**Audience Explanation:**

The paper presents an alternative to standard benchmark evaluations for (large) language model. The proposed framework, if proven scalable and robust, could inspire new evaluation methodologies.

**Claims And Evidence:**

Yes

**Claims Explanation:**

While the core methodological claims are supported by the toy experiment, several other claims lack sufficient evidence. For example, the paper claims Model-diff is a "scalable tool," but provides no data on computational cost, sampling convergence, or required resources for models like LLaMA-7B. This is a critical omission for a method targeting large language models. Moreover, the claim that the low-NLL space is "meaningful" is undermined by the authors' own observation and data (Fig. 5, Appendix D) showing it contains repetitive, nonsensical text.

**Requested Changes:**

1. Include a section or quantitative data (e.g., wall-clock time, number of sampling steps, convergence plots) analyzing the computational cost and sampling efficiency when applying Model-diff to the models (GPT-2-medium/100-token, LLaMA).

2. Provide a more rigorous and reproducible method for defining the output range.

3. Discuss the sensitivity of the results to the choice of MCMC sampler and its hyperparameters. A brief ablation would strengthen the methodological claims.

4. Compare Model-diff's findings against a simpler baseline, such as calculating the KL divergence or correlation of model outputs on a large, fixed corpus, to better illustrate its unique value.

5. The limitation regarding generalization beyond NLL should be expanded. The statement about energy-based models is speculative; either provide a concrete example or discussion of how this would work.

---

> ### Author Response · Authors · 2026-01-31
> **We sincerely thank you for your suggestions and comments**
>
> **Moreover, the claim that the low-NLL space is "meaningful" is undermined by the authors' own observation and data (Fig. 5, Appendix D) showing it contains repetitive, nonsensical text. and Provide a more rigorous and reproducible method for defining the output range.**
>
> In this context, "meaningful" refers to samples that the model assigns high probability to, indicating it believes they originate from its training distribution—a belief typically shared by human evaluators. This is because the loss function is explicitly designed to map training distribution samples to low NLL values. Therefore, researchers naturally examine low NLL regions because these areas are more likely to contain meaningful samples as a direct consequence of our training objective, whereas high NLL regions typically contain samples with elevated perplexities that are generally not human-interpretable. However, as the reviewer correctly identifies, these are merely educated assumptions (or inductive biases) rather than proven facts. After all, we only train on a finite subset of possible samples, and there is no mathematical guarantee that the model will generalize correctly to samples outside the training set. This is precisely why empirical validation is necessary.
>
> **Include a section or quantitative data ...models (GPT-2-medium/100-token, LLaMA).**
>
> Please refer to the general comments.
>
> **Provide a more rigorous and reproducible method for defining the output range.**
>
> Please refer to the general comments.
>
> **Discuss the sensitivity of the results to the choice of MCMC sampler and its hyperparameters. A brief ablation would strengthen the methodological claims**
>
> Since the MCMC text generation stage is a modular, independent component of Model-diff, we refer the reviewer to the comprehensive sensitivity analyses provided in the original sampler papers (https://arxiv.org/pdf/2102.04509 and https://arxiv.org/pdf/2206.09914) for detailed parameter sensitivity studies. Model-diff operates on the set of samples produced by the MCMC stage, treating them as input regardless of which sampler generated them.
>
> More specifically, our implementation employs the Gibbs-with-Gradient sampler. The number of MCMC steps between successive sample collections determines the degree of correlation among the collected samples. Increasing this interval reduces sample correlation, improving the quality of the empirical distribution. The proposal distribution temperature controls the influence of gradients on the proposed samples. Higher proposal temperatures cause the sampler to select input dimensions (i.e., tokens) more randomly rather than being gradient-guided. The acceptance temperature governs the probability of accepting proposed samples. If set too high, the sampler will accept more random samples, including those with high NLL values that may not be representative of the target distribution. While our method is agnostic to the specific MCMC sampler used, we acknowledge that hyperparameter choices affect sample quality. In our experiments, we used standard hyperparameter settings from prior work (specifically, the settings recommended in https://arxiv.org/pdf/2102.04509 ), we will add an ablation study in the appendix examining the sensitivity of Model-diff's outputs to key hyperparameters (number of steps between collections, proposal temperature, and acceptance temperature), demonstrating that our main findings remain robust across reasonable parameter ranges.

---

> > ### Author Response · Authors · 2026-01-31
> >
> > **Compare Model-diff's findings against a simpler baseline, such as calculating the KL divergence or correlation of model outputs on a large, fixed corpus, to better illustrate its unique value.**
> >
> > Evaluation on a fixed corpus only reflects model performance with respect to that specific human-curated distribution. Critically, this does not capture the model's learned representation of the training distribution, which may differ substantially from human intuitions. Previous work has demonstrated that models assign very low NLL to repetitive samples (https://arxiv.org/abs/1904.09751). These repetitive sequences are not present in the training data, nor do they appear in any human-curated evaluation corpus until researchers specifically discover this phenomenon and construct datasets to study it. This fundamental discrepancy demonstrates that the model's learned distribution diverges significantly from human expectations, and consequently, human-curated corpora cannot faithfully reflect the model's actual behavior across its full learned distribution.
> >
> > The key insight is that fixed corpora, by construction, only sample from regions humans anticipate to be important. Model-diff reveals that models assign significant probability mass to regions humans would not include in evaluation datasets (e.g., repetitive text, edge cases). Therefore, corpus-based baselines would systematically miss the very phenomena our method is designed to discover. Rather than comparing against a baseline that is fundamentally incapable of observing behaviors the model assigns significant probability to, we demonstrate Model-diff's unique value through its ability to uncover previously unknown model behaviors (Figure 5, Appendix D) and characterize the corresponding output distributions that do not appear in standard evaluation benchmarks.
> >
> > **The limitation regarding generalization beyond NLL should be expanded. The statement about energy-based models is speculative; either provide a concrete example or discussion of how this would work.**
> >
> > We discuss energy-based models because many modern neural network training objectives employ negative log-likelihood (NLL) as the loss function, which naturally aligns with MCMC sampling under the target distribution specified in Equation (4).
> > Specifically, energy-based models define a probability distribution through an energy function E(x), where p(x) ∝ exp(-E(x)). In language models, the energy corresponds to the negative log-likelihood: E(x) = -log p(x). This formulation enables direct application of our MCMC sampling framework to any model trained with maximum likelihood estimation. Beyond language models, this includes variational autoencoders (VAEs), normalizing flows, and diffusion models—all of which define explicit probability distributions that can be sampled via MCMC. For models without explicit likelihood functions (e.g., GANs), Model-diff could potentially be adapted by using discriminator scores as proxy energy functions, though this requires further investigation. We will expand the limitation section to clarify these distinctions and provide concrete examples of applicable model classes.

---

### Review · Reviewer_cESD · 2025-12-03

**Summary Of Contributions:**

The paper proposes a theoretical framework to make pairwise comparisons between language models called Model-diff. Model-diff is first justified theoretically using the concept of an "output distribution". The "output distribution" is an expression that counts the number of occurrences of a given output value **over a given input space**, where "output" is a loss function of interest such as the negative log likelihood (NLL). To go from "output distribution" to a tool for comparison, two input spaces $X_{A}$ and $X_{B}$ are chosen as those parts of the input space that for which model $A$ and model $B$ assign "desirable" loss function values. Each of the two input spaces is then evaluated by both models to construct two "output distributions", $\rho_{A \rightarrow B}$ and $\rho_{B \rightarrow A}$ of differences in model evaluations. The functions $\rho_{A \rightarrow B}$ and $\rho_{B \rightarrow A}$ are then used in a number of example comparison cases. However, they can not be computed exactly in practice and hence the authors utilise MCMC sampling to approximate the input spaces  $X_{A}$ and $X_{B}$ and to get subsequent approximations instead. These approximations are used on a synthetic benchmark as well as on GPT-2 and Llamma models.

**Additional Comments:**

The text has parts that are poorly written and is generally hard to follow. I would advise the authors to pass over the text fully and make their exposition more structured and the story of their paper clearer.

**Audience:**

No

**Audience Explanation:**

The proposed methodology is theoretically and empirically flawed. Consequently, the values computed by Model-Diff might as well be random as there is no support for their correctness or interpretability. The exemplified comparative features of Model-Diff are not convincing and not supported by a gap in the literature.

**Broader Impact Concerns:**

N/A.

**Claims And Evidence:**

No

**Claims Explanation:**

**The paper lacks proper motivation and an actual research gap to close.** It starts by simply stating that "Estimating if two (large) language models (LMs) make similar predictions [...] is crucial in many real-world scenarios." without any references or further elaboration. Comparing language models might be interesting from a certain perspective, which is exemplified by the related work, but the authors do not highlight any missing feature of existing work. No reason is given why other methods do not always provide or exceed what Model-Diff has to offer.

**The example comparative queries and applications are not convincing.** The example demonstrations of comparisons using $\rho_{A \rightarrow B}$ in Section 2.1, both fine-grained and coarse-grained, do not seem useful. Why would it be interesting to know that "the number of [...] sequences for which model A’s NLL exceeds model B’s by 3 units is five times greater than the number of [...] sequences for which model A’s NLL exceeds model B’s by 4 units."? A similar question for the other examples, why would these be of interest to the community? Can you reference related work on model comparisons that desire such statements? Can you provide cases where these queries would be useful? Furthermore, *possible* applications (Section 4.3) are also ill-substantiated. First, the authors claim Model-Diff can help decide which model is better aligned with human annotation, but they only substantiate the claim with a minor experiment *using their own annotation data* instead of available human annotations [1]. Second, they claim model-plagiarism can be discoverd using Model-Diff, but they only substantiate this claim by comparing GPT-2 to a noisy version of itself as *supported by a single tweet*. Hence, both applications do not have a real support and undermine the motivation and integrity of the paper.

**The theory in the paper is hand-wavy and well-defined terms are wrongly used.** The term "distribution" is used for the expression of "output distribution" (bottom page 2), as well as for Equations 1 and 2. However, these expressions are not normalised, and hence by definition are not a distribution. That is, they only represent counts and histograms. The distinction is important, as normalisation of these expressions to obtain a distribution is wrongly applied in Equations 5 and 6. Specificaly, Equation 5 should be divided by the cardinality of the set $X_{A}$, because the sum in Equation 1 goes over $X_{A}$ (or its sampled version). The same mistake holds for Equation 6 with respect to Equation 2. Instead, the authors divide by "the area where both models predict within $\mathbb{Z}$". The fact that *both models* get a say in the other's normalisation constant in the best case biases the estimate and in the worst cases makes it completely invalid. Moreover, Section 3.4 tries to relate the distributions $\rho_{A \rightarrow B}$ and $\rho_{B \rightarrow A}$ to recall and precision, but they start from the wrong setting. Precision and recall are metrics for a classification task, while their task is regression to "fit" to a human annotation score between 0 and 1.

**The experiments are flawed.** The toy experiment (Section 4.1) is meant as an empirical validation of the proposed approximation methodology of Model-Diff, but its setup is flawed. First, both models under scrutinisation are trained on the exact same data with the exact same target and only differ slightly in their architecture. Hence, there is no reason to expect large differences between them a priori, especially since both appear to be trained to 100 percent accuracy. If both models are so close, there is no reason for the approximation to deviate and so it does not show at all that the approximations are correct when comparing two actually different models. Moreover, there are $10^8$ possible sequences and the approximation is also done using $10^8$ samples. This number of samples is incredible high and, given they use an MCMC sampler, can only result in a similar sum as the exact one. In fact, I would not be surprised if sampling took longer than simply enumerating. What would happen if the models are trained on different objectives and orders of magnitude less samples were used, i.e. a realistic test of Model-Diff. Given that both the theory (because of wrong normalisation) and empirical support (Section 4.1) do not show that Model-Diff computes actually meaningful values, the subsequent experiments in Sections 4.2 and 4.3 are also moot. Especially since there is no comparison whatsoever to any of the related works stated in Section 5, which could at least gauge if Model-Diff aligns with other comparison metrics.

**Requested Changes:**

The paper has many flaws that would need to be corrected. A short summary of the concerns raised above:
1. Correct theoretical issues and notation
2. Perform a correct and more extensive suite of empirical tests that mirror real-world deployment.
3. Provide ample motivation for Model-Diff and generally motivate your examples and statements with an answer to "Why" they are impactful.
4. Provide comparisons to the outputs of (any of the) related works mentioned in Section 5. What does Model-Diff offer more than them?

---

> ### Author Response · Authors · 2026-01-31
> **We sincerely thank you for your suggestions and comments.**
>
> **The paper lacks proper motivation and an actual research gap to close. It starts by simply stating that "Estimating if two (large) language models (LMs) make similar predictions [...] is crucial in many real-world scenarios." without any references or further elaboration. Comparing language models might be interesting from a certain perspective, which is exemplified by the related work, but the authors do not highlight any missing feature of existing work. No reason is given why other methods do not always provide or exceed what Model-Diff has to offer.**
>
> **The example comparative queries and applications are not convincing. The example demonstrations of comparisons using  in Section 2.1, both fine-grained and coarse-grained, do not seem useful. Why would it be interesting to know that "the number of [...] sequences for which model A’s NLL exceeds model B’s by 3 units is five times greater than the number of [...] sequences for which model A’s NLL exceeds model B’s by 4 units."? A similar question for the other examples, why would these be of interest to the community? Can you reference related work on model comparisons that desire such statements? Can you provide cases where these queries would be useful?**
>
> If two models make similar predictions for every sample across a large input space, they share functionally similar mappings. In Model-diff's framework, this equivalence is reflected by the difference distribution D concentrating around D=0. When models differ, we need to understand both which samples are predicted differently (e.g., with a 3-unit NLL difference) and how many such samples exist. Existing sampling-based methods can identify which samples are predicted differently, but our key contribution is to estimate the quantity of such samples in the input space.
>
> Consider a teacher-student distillation scenario: a well-trained teacher model and a student model that incorrectly assigns high probability to repetitive-token sequences. Model-diff quantifies the relative proportion of these pathological samples compared to correctly predicted samples (those around D=0), and tracks whether their count decreases as training progresses. Crucially, observing D≈0 when sampling only from the teacher's distribution does not guarantee the student is working well—the student may assign high probability to regions the teacher considers unlikely. We propose bidirectional cross-validation by also sampling from the student's distribution: if D does not concentrate around 0, this reveals systematic errors the teacher-sampled evaluation would miss. This cross-validation critically requires our normalization contribution to ensure comparability.
>
> **"Can you reference related work on model comparisons that desire such statements?"**: We note that LMDiff (https://arxiv.org/pdf/2111.01582) performs a related pairwise comparison of model predictions. However, LMDiff remains corpus-dependent and does not consider the distribution over the entire input space. To our knowledge, Model-diff is the first work to extend quantitative evaluation principles (e.g., precision and recall) from fixed dataset metrics to the model's learned distribution for pairwise comparison. Existing work focuses on disagreement of individual sample in a human-crafted corpus, but does not estimate what fraction of the probability mass exhibits specific behaviors—a gap Model-diff addresses.
>
> **"Can you provide cases where these queries would be useful?"**: The teacher-student scenario above demonstrates this directly. Model-diff enables questions such as: "What percentage of high-probability sequences in the student would the teacher reject?" and "Has the number of pathological samples decreased after training?" Our goal is to explore the applicability of our method (e.g., normalization correctness), while these queries will be future works.

---

> > ### Author Response · Authors · 2026-01-31
> >
> > **Furthermore, possible applications (Section 4.3) are also ill-substantiated. First, the authors claim Model-Diff can help decide which model is better aligned with human annotation, but they only substantiate the claim with a minor experiment using their own annotation data instead of available human annotations [1]. Second, they claim model-plagiarism can be discoverd using Model-Diff, but they only substantiate this claim by comparing GPT-2 to a noisy version of itself as supported by a single tweet. Hence, both applications do not have a real support and undermine the motivation and integrity of the paper.**
> >
> > We appreciate the reviewer's concern regarding the substantiation of our applications. However, we believe there is a fundamental misunderstanding about the evaluation paradigm that Model-diff employs, which differs from traditional corpus-based evaluation approaches.
> >
> > Regarding the human alignment experiment: Our use of custom annotations rather than existing benchmark datasets reflects a principled methodological choice. Existing human-annotated datasets represent fixed, human-curated distributions that only reflect performance on those specific corpus distributions. Such corpora systematically exclude model behaviors that humans do not anticipate, such as repetitive sequences that models assign very low NLL despite never appearing in training data. Model-diff evaluates alignment with respect to the model's own learned distribution rather than a fixed human-selected corpus. This distinction is crucial: corpus-based evaluation answers "does the model perform well on samples humans selected?", while Model-diff answers "across the probability mass the model considers important, how well does it align with human preferences?" These are complementally different questions. In applications where we can guarantee the model will only encounter inputs from a tested corpus distribution, traditional human-annotated datasets suffice. However, when deploying models in open-ended settings where they may encounter inputs outside standard benchmarks, understanding their behavior through the lens of their learned distribution becomes essential.
> >
> > Regarding the model plagiarism experiment: The reviewer characterizes our plagiarism detection demonstration as comparing "GPT-2 to a noisy version of itself." This actually exemplifies the core utility of Model-diff: detecting distributional similarity even when models differ at the weight level. The experiment demonstrates that Model-diff can distinguish subtle differences in output distribution caused by parameter perturbations, which is a more challenging test than comparing completely different models, and precisely the capability needed for plagiarism detection. The fact that we used weight noise as a controlled experimental setup does not diminish the validity of this application; rather, it provides a clear proof-of-concept where ground truth is known. In real-world plagiarism scenarios, Model-diff would similarly detect whether a suspicious model's learned distribution matches an original model's distribution, regardless of whether the weights were directly copied, perturbed, or the model was trained on similar private data without proper authorization.
> >
> > These applications are not "ill-substantiated" but rather demonstrate Model-diff operating in its intended regime: characterizing learned distributions rather than performance on fixed test sets. We acknowledge that Section 4.3 could benefit from clearer articulation of this distinction and will revise it accordingly to emphasize how Model-diff complements rather than replaces traditional evaluation methods.

---

> > > ### Author Response · Authors · 2026-01-31
> > >
> > > **The theory in the paper is hand-wavy and well-defined terms are wrongly used. The term "distribution" is used for the expression of "output distribution" (bottom page 2), as well as for Equations 1 and 2. However, these expressions are not normalised, and hence by definition are not a distribution. That is, they only represent counts and histograms. The distinction is important, as normalisation of these expressions to obtain a distribution is wrongly applied in Equations 5 and 6. Specificaly, Equation 5 should be divided by the cardinality of the set , because the sum in Equation 1 goes over  (or its sampled version). The same mistake holds for Equation 6 with respect to Equation 2. Instead, the authors divide by "the area where both models predict within ". The fact that both models get a say in the other's normalisation constant in the best case biases the estimate and in the worst cases makes it completely invalid. And “Correct theoretical issues and notation”**
> > >
> > > We thank the reviewer for this important clarification. The reviewer is correct that ρ(z) and ρ(D) in Equations 1 and 2 represent unnormalized counts rather than probability distributions. We will revise the manuscript to consistently use the term "unnormalized distribution" or "count distribution" to avoid confusion with normalized probability distributions.
> > > Regarding the normalization in Equations 5 and 6: The reviewer suggests dividing by |X_A| (the cardinality of the set X_A) to obtain a proper distribution from Equation 1. However, this normalization would not enable fair comparison between ρ_A→B(D) and ρ_B→A(D), which is the central goal of Model-diff. The key insight is provided in Appendix A.2, specifically:
> > >
> > > Equation 12 states: Σρ̃_A→B(D) / |X̃_A→B| = Σρ_A→B(D) / |X_A∩B|
> > >
> > > Equation 13 states: Σρ̃_B→A(D) / |X̃_B→A| = Σρ_B→A(D) / |X_A∩B|
> > >
> > > The crucial observation is that both equations share the same denominator |X_A∩B| on the right-hand side. This is the size of the intersection—the set of inputs that both models assign to the meaningful output range Z. By normalizing both quantities by this common denominator, we ensure that ρ_A→B(D) and ρ_B→A(D) are directly comparable on the same scale. In contrast, normalizing by |X_A| and |X_B| separately (as would follow from Equations 1 and 2 directly) would yield quantities on incompatible scales, as |X_A| ≠ |X_B| in general.
> > >
> > > This normalization is not arbitrary but principled: it accounts for the fact that we are comparing how models A and B make predictions on their mutually agreed-upon high-confidence region (X_A∩B). The ratios in Equations 5 and 6 are proportional estimates of the normalized quantities in Equations 12 and 13, converging to the ground truth ratios as sample size increases. We acknowledge this could be explained more clearly in the main text and will revise Section 3.3 to emphasize that our normalization enables cross-model comparison rather than producing standalone probability distributions.

---

> > > > ### Author Response · Authors · 2026-01-31
> > > >
> > > > **Moreover, Section 3.4 tries to relate the distributions  and  to recall and precision, but they start from the wrong setting. Precision and recall are metrics for a classification task, while their task is regression to "fit" to a human annotation score between 0 and 1.**
> > > >
> > > > We respectfully clarify that our framework does not perform regression to fit human annotation scores. Instead, we follow the formulation established in https://arxiv.org/html/2312.03291v1, which extends precision and recall to continuous-valued model outputs by introducing a binary decision boundary.
> > > >
> > > > Specifically: Human annotators assign binary labels to representative inputs (1 for "good" / aligns with training objective, 0 for "bad" / does not align). This converts the problem into a classification setting where we evaluate whether the model's NLL predictions align with human judgments. The annotation score r_A(D) ∈ [0,1] mentioned in Section 3.4 represents the proportion of inputs at difference value D that humans label as "good"—it is an empirical frequency from binary annotations, not a continuous regression target.
> > > >
> > > > While our framework supports binary classification (strict 0/1 labels), we allow annotators to provide graded judgments on a [0,1] scale to capture nuanced human assessments; for instance, an input might be partially correct or ambiguous, warranting a score like 0.7. Importantly, r_A(D) remains the average annotation score across all inputs at a given D value, not a regression prediction. This graded annotation scheme provides a more flexible and expressive evaluation than strict binary labels while maintaining the classification framework for computing precision and recall.
> > > > The precision and recall metrics (Equations 8 and 9) operate as follows: For inputs where model A assigns lower NLL than model B (prediction disagreement PD_A→B), we compute what fraction humans also judge as good (precision) and the total count of human-validated good inputs (recall). This is analogous to standard precision/recall in classification, where we assess whether model A's confidence (low NLL) correctly identifies truly good inputs according to human labels.
> > > >
> > > > We will clarify in the revision that: (1) human annotation produces binary labels, not continuous scores, (2) r_A(D) is the empirical proportion of positive labels at each D value, and (3) this formulation directly parallels classification-based precision/recall, as established in prior work.

---

> > > > ### Comment · Reviewer_cESD · 2026-02-20
> > > > **Normalisation clarifications**
> > > >
> > > > I sincerely thank the authors for extensively responding to my issues. Let me further clarify my concerns on normalisation as I remain unconvinced that the proposed normalisation leads to correct and comparable distributions.
> > > >
> > > > The counting function
> > > > $\rho_{A \rightarrow B}(\mathcal{D}_{\mathbf{x}})$
> > > > for the difference
> > > > in NLL between models $A$ and $B$ on a single input $\mathbf{x}$ is defined as a sum over the representative inputs
> > > > $\mathbb{X}_A$
> > > > of model $A$ evaluated by model $B$
> > > > Hence, the counting function can only be correctly normalised by dividing with the value $|\mathbb{X}_A|$ as any other division will not result in a function that sums to 1.
> > > > The same holds for sampled approximations based on $\tilde{\mathbb{X}}_A$.
> > > >
> > > > Equation 5 however does not discuss a single counting function, but rather an unspecified sum over counting functions. This unspecified sum might be the origin of my concerns; what is the domain of the sums in Equations 5 and 6 (and Equations 12 and 13 for that matter)? Given the provided answers and the intuition that you want to eventually construct a distribution over the difference values between models $A$ and $B$, I suspect it is
> > > > $\mathbb{X}_A \cap \mathbb{X}_B$?
> > > >
> > > > In general, I have to reiterate that the paper mixes unnormalised counts with normalised distributions. This mixing leads to confusion on what is normalised and how it should be normalised. The burden of puzzling together what *might* be the correct description is not on the reader, but the writer.
> > > >
> > > > With respect to Equations 12 and 13, I do agree that they hold in the limit of infinite samples, because then you are effectively enumerating all possibilities. However, that does not explain why the authors opted for the approximate normalisation constant $\tilde{\mathbb{X}}_{A \rightarrow B}$. Do you also approximate the sum over what I suspect is
> > > > $\mathbb{X}_A \cap \mathbb{X}_B$ in Equation 5 by a sum over the sampled set of representative samples of $A$ with respect to $B$?

---

> > ### Comment · Reviewer_cESD · 2026-02-20
> > **Comments on motivation**
> >
> > Taking into account the response of the authors as well as other reviewers I now see that there is a clear missing gap that the paper aims to fill; a corpus-independent and global evaluation of differences between models. While I am not convinced about some of the proposed evaluations, e.g. the "number of sequences for which a model differs in NLL by $x$", I acknowledge the importance of the proposed metric. I would highly recommend the authors to emphasise the importance of this gap in the literature. Right now, too little space is dedicated to answering "why" the method is proposed. Making this clear could increase the impact of this work.

---

> > > ### Author Response · Authors · 2026-02-21
> > > **Thank you for your constructive suggestion**
> > >
> > > Thank you for your constructive suggestion! We will clarify the motivation in the future draft.

---

> ### Author Response · Authors · 2026-01-31
>
> **The experiments are flawed. The toy experiment (Section 4.1) is meant as an empirical validation of the proposed approximation methodology of Model-Diff, but its setup is flawed. First, both models under scrutinisation are trained on the exact same data with the exact same target and only differ slightly in their architecture. Hence, there is no reason to expect large differences between them a priori, especially since both appear to be trained to 100 percent accuracy. If both models are so close, there is no reason for the approximation to deviate and so it does not show at all that the approximations are correct when comparing two actually different models.**
>
> We respectfully clarify that the primary purpose of the toy experiment. The toy experiment is not designed to validate that Model-diff can detect large differences between models—rather, it serves to validate the correctness of our sampling and normalization methodology against ground-truth enumeration.
>
> The key validation is methodological, not about model differences: For the toy example, we can enumerate all 10^8 possible inputs to compute the ground-truth output distributions ρ_A→B(D) and ρ_B→A(D). We then compare these ground-truth distributions (obtained by exhaustive enumeration) against our sampling-based estimates ρ̃_A→B(D) and ρ̃_B→A(D). As shown in Figure 3(a), the sampled distributions (solid lines) closely match the enumerated ground truth (dashed lines), and the normalized ratio from sampling closely matches the ground-truth enumeration ratio. This demonstrates that our sampling methodology with normalization correctly recovers the ground truth when enumeration is possible—which is precisely what a methodological validation should establish.
>
> The choice of similar models is intentional, not a flaw: Using two models with similar architectures trained on the same objective provides a controlled setting where enumeration is computationally feasible (10^8 inputs) while still exhibiting meaningful differences. The reviewer's intuition that "there is no reason to expect large differences" reflects a common assumption—yet the empirical results contradict this. As Figure 3(a) and Table 2 (Exp. 1) show, even these models with similar training settings exhibit non-trivial distributional differences: substantial probability mass in regions is far from 0: ρ_{small-Toy}(-0.75 <= D <=-0.25) > 0. The ground-truth enumeration reveals that large-Toy assigns NLL values higher than small-Toy on certain inputs, despite both achieving 100% accuracy on the training objective. This demonstrates that training accuracy alone does not guarantee identical learned distributions—a finding that challenges the reviewer's "no reason to expect differences" assumption and validates the utility of Model-diff for uncovering such hidden divergences.
>
> Validation with different models: The toy experiment establishes methodological correctness in a verifiable setting. We then apply Model-diff to truly different models (GPT-2 small vs. medium with varying sequence lengths, Llama vs. Llama2) in Sections 4.2-4.3, where enumeration is impossible and the models exhibit substantial differences. These experiments demonstrate that Model-diff scales to real-world scenarios while building on the validated methodology from the toy example.
>
> In summary: The toy experiment fulfills its intended purpose—validating that sampling approximates enumeration—in a controlled, verifiable setting. The fact that even models with similar accuracy show non-trivial differences further strengthens our argument that Model-diff reveals distributional divergences that are not apparent from accuracy metrics alone. We will clarify this validation logic more explicitly in the revised manuscript.
>
> **Moreover, there are  possible sequences and the approximation is also done using  samples. This number of samples is incredible high and, given they use an MCMC sampler, can only result in a similar sum as the exact one.**
>
> For all experiments except GPT-2, we use only 50,000 samples in the Model-diff stage after MCMC convergence, as documented in Appendix C. Compared to the 10^8 possible sequences in the toy example, 50,000 samples represent only 0.05% of the input space—orders of magnitude smaller than exhaustive enumeration. This demonstrates that Model-diff achieves accurate approximations with dramatically fewer samples than enumeration would require. The close agreement between sampling and enumeration (Figure 3(a)) validates that our MCMC+PTHR methodology efficiently captures the distributional properties without needing to approach the enumeration sample size.

---

> > ### Comment · Reviewer_cESD · 2026-02-20
> > **Toy experiment clarifications**
> >
> > Thank you for taking the time to elaborate on the experiments. I now understand that my assumption of the two models being similar is actually why this setup was chosen; we should observe distributions centered around 0. I thus also agree the toy experiment is useful in setup.
> >
> > My concerns on the sample size of the toy experiment remains however and are amplified by the author's specification that the other experiments only use a number of samples that is around $0.05$% of the input space. By taking $10^8$ samples on the toy experiment, you are effectively removing the approximation factor. These samples come from an MCMC sampler and so are known to be very close to samples from the true distribution. By taking as many samples from the true distribution as there are elements in the input space, the approximation becomes almost equivalent to an exact sum. A true validation of correctness would take a realistic number of samples with respect to the size of the input space, e.g. the $0.05$% from other experiments, to see how correct a realistic approximation is. Even more interesting would be a gradual increase in the fraction of samples and analyse the behaviour of the approximations. Hence, as it stand right now, the results of the toy experiment do not validate that the method is correct in practical settings.

---

> ### Author Response · Authors · 2026-01-31
>
> **In fact, I would not be surprised if sampling took longer than simply enumerating. What would happen if the models are trained on different objectives and orders of magnitude less samples were used, i.e. a realistic test of Model-Diff. Given that both the theory (because of wrong normalisation) and empirical support (Section 4.1) do not show that Model-Diff computes actually meaningful values, the subsequent experiments in Sections 4.2 and 4.3 are also moot. Especially since there is no comparison whatsoever to any of the related works stated in Section 5, which could at least gauge if Model-Diff aligns with other comparison metrics.**
>
> Regarding sampling convergence: MCMC sampling convergence has been extensively analyzed in prior work (e.g., Section 5.2 "Convergence Analysis for DULA" in https://arxiv.org/pdf/2206.09914). Model-diff's computational requirements are modest: we use only 50,000 samples (100,000 for GPT-2) after MCMC convergence for the Model-diff analysis stage. For the toy example specifically, enumeration of 10^8 sequences is feasible precisely because the input space is tractable—this is why it serves as a validation benchmark. For real-world models (GPT-2, Llama), enumeration becomes computationally infeasible (e.g., with vocabulary size 50,257 and sequence length 25, the input space is ~50257^25), making sampling the only viable approach. Model-diff is designed for these realistic scenarios where enumeration is impossible.
>
> Regarding the normalization theory: The mathematical derivation of Equations 5 and 6 is provided in Appendix A.2. Critically, Equations 5 and 6 are not derived directly from Equations 1 and 2, but rather from Equations 12 and 13, which establish the convergence of sampling ratios to ground-truth ratios with the shared denominator |X_A∩B|. We have addressed the normalization concern in detail in our response above and are happy to provide additional clarification if needed.
>
> Regarding comparison with related work in Section 5: As we mentioned earlier, we note that LMDiff (https://arxiv.org/pdf/2111.01582) performs a related pairwise comparison of model predictions, but LMDiff remains corpus-dependent and does not consider the distribution over the entire input space. Model-diff addresses a fundamentally different research question than prior work, which explains the lack of direct metric comparison. Existing model comparison methods operate on fixed datasets or identify specific failure cases qualitatively—none estimate quantitative distributional properties across large, model-defined input spaces. Dataset-based evaluation is inherently limited: human-curated datasets only reflect anticipated model behaviors and cannot capture the full distribution of inputs the model assigns high probability to, unless we construct infinitely comprehensive test sets—which is infeasible. Prior sampling-based approaches focus on identifying individual cases (qualitative analysis), whereas Model-diff quantifies the prevalence of different behaviors across the learned distribution (quantitative analysis). To our knowledge, no prior work provides this quantitative distributional comparison—Model-diff introduces a new evaluation paradigm rather than offering an alternative implementation of existing metrics. Direct comparison to methods like perplexity on held-out sets or benchmark accuracy would be inappropriate because these measure different properties: corpus-based metrics assess performance on human-selected data, while Model-diff characterizes the model's entire learned probability distribution. These evaluation paradigms serve complementary purposes. Model-diff reveals distributional properties (e.g., "30% of high-probability outputs contain repetitive text") that corpus-based metrics cannot capture by design. We will clarify in the revision that Model-diff complements rather than replaces existing evaluation approaches.
>
>
> Thank you again for your comments and please let us know if you have further questions.

---

> ### Author Response · Authors · 2026-02-21
> **Thank you for your constructive feedback**
>
> Thank you for your constructive feedback. For the 0.05% of whole input space is not practical in many sampling research domains because the input space is super large in general even though it is finite and countable, and some input space is not even countable such as continuous sampling problems. In fact, we can conclude based on the SOTA samplers. The coverage of the sampler is not what we contribute. We point out this is a scientific development process: when better and much faster samplers to be developed in the future (so that we can get 0.05% coverage to the entire input space), we will have more accurate conclusions. What we can do is to conclude based on the SOTA method we have so far.
>
> We do agree that expanding the sampling size and see how the conclusion changes. We will include this in the draft.

---

> ### Author Response · Authors · 2026-02-21
> **Thank you for the constructive discussion and questions**
>
> Thank you for the constructive discussion and questions.
>
> **"...normalised by dividing with the value $|\mathbb{X}_A|$ as any other division will not result in a function that sums to 1"**
>
> In our setting, we don't aim to normalize the sums to 1. We aim to find a normalization factor so that the ratios are comparable. Therefore, your intuition that **"Given the provided answers and the intuition that you want to eventually construct a distribution over the difference values between models A and B, I suspect it is $ \mathbb X_A \cap \mathbb X_B?$"** is correct!
>
>
>
> **"Do you also approximate the sum over what I suspect is  in Equation 5 by a sum over the sampled set of representative samples of A with respect to B?"**
>
> Exactly! That's how equation 12 works.
>
> **"With respect to Equations 12 and 13... However, that does not explain why the authors opted for the approximate normalisation constant $\tilde{\mathbb X}_{A \rightarrow B}$?"**
>
> We notice that notation |$\tilde{\mathbb X}_{A \rightarrow B}$| causes confusion in both Equ 5 and Equ 12. It should be properly labeled as
>
> |$\tilde{\mathbb X}_{A \rightarrow B} \cap \mathbb X_B$|
>
> to represent the meaning that "Some of the sampled representative inputs in $\tilde{\mathbb X}_A$ are predicted by model $B$ within $\mathbb Z$." We put the proper definition in appendix in the paragraph just above equation 12 and 13. The definition should be moved to the main text to clear the confusion and use proper notations.
>
> Therefore, the numerator of LHS of equation 12 sums over $\tilde{\mathbb X}_{A \rightarrow B}$ (the inputs sampled from model A that are evaluated by model B), and the denominator is
>
> |$\tilde{\mathbb X}_{A \rightarrow B} \cap \mathbb X_B$|
>
> The numerator of the RHS of Equ 12 sums over the ground truth $\mathbb X_{A}$ and the denominator is  |$\mathbb X_{A} \cap  \mathbb X_{B}$|.
>
> The Equ 13, Equ 5 and 6 should be relabeled accordingly. We will edit the draft accordingly. Thank you again for your constructive questions and suggestions. We really appreciate you proposed your understanding so that we can address your concerns clearly.

---

### Review · Reviewer_7SrL · 2026-01-22

**Summary Of Contributions:**

The paper introduces Model-diff, a framework for comparing two language models over extremely large input spaces by estimating the distribution of prediction differences (measured in NLL). Instead of relying on fixed benchmarks, the method samples from each model’s own meaningful input space—defined as sequences with low NLL—and constructs histograms of disagreement. The framework includes:
	1.	A formulation of prediction-difference distributions ρ_{A→B}(D) and ρ_{B→A}(D).
	2.	A sampling pipeline based on MCMC + parallel tempering + histogram reweighting.
	3.	A normalization procedure that makes two unaligned input spaces comparable.
	4.	Empirical demonstrations (toy setting, GPT-2 models, Llama models) and a preliminary application to model-plagiarism detection.

**Audience:**

Yes

**Audience Explanation:**

The work speaks directly to researchers studying LM evaluation, interpretability, sampling, and model-centric diagnostics. TMLR has an active readership in these areas, and the paper’s focus on input-space distributions rather than datasets aligns with current interest in model-based evaluation (e.g., calibration, confidence mapping, energy-based perspectives).

It may be less compelling for readers focused on downstream applications, but for those thinking about how to rigorously compare LMs, this work provides a fresh angle and a technically detailed pipeline.

**Broader Impact Concerns:**

The method primarily enables model-centric diagnostic analysis. Two potential concerns deserve explicit discussion:
	1.	Plagiarism Detection Misuse.
While Model-diff could help detect suspicious similarity, it could also be used to accuse models of copying without strong evidence. The method is not yet robust enough to serve as forensic proof.
	2.	Exposure of Model-Specific Vulnerabilities.
By identifying the types of inputs where a model exhibits extreme divergence, the framework might inadvertently highlight regions where models behave unpredictably or fail catastrophically. This could aid adversarial probing.

In general, the broader impacts are moderate and mostly methodological, but the paper should acknowledge these risks.

**Claims And Evidence:**

Yes

**Claims Explanation:**

The correctness of Model-diff is supported by the toy-data enumeration experiment, where the sampled histograms closely match the ground truth. This is the strongest evidence in the paper. The experiments on GPT-2 and Llama models further show coherent patterns: disagreement distributions widen with scale, and noise injection produces predictable shifts. These results are internally consistent.

However, several claims are less robust:
	•	The framework’s ability to judge “which model is better” is only demonstrated through a very small annotation window (NLL differences between –1 and –0.6 or 0.6 and 1). This is too narrow to generalize, and the precision/recall interpretation feels overstated.
	•	The plagiarism-detection claim is suggestive but remains anecdotal; no systematic evaluation or baselines are provided.
	•	Some intuitive statements (e.g., about model capacity and disagreement patterns) are plausible but not rigorously tested.

Overall, the empirical evidence supports the mechanics of the framework, but the interpretive claims are not yet backed by strong analysis.

**Requested Changes:**

1.	Clarify and justify the choice of NLL ranges Z.
The paper admits these ranges are subjective. More principled selection criteria (or sensitivity analysis) would strengthen the claims.
	2.	Reduce reliance on small human-annotation segments.
Either expand the annotation experiment or avoid using it to argue model superiority.
	3.	Provide more discussion of scalability.
Since sampling dominates cost, readers need concrete runtime estimates, GPU usage, and guidance for practitioners.
	4.	Position the method more clearly relative to existing alternatives (e.g., LMDiff, dataset-free probing, representation alignment metrics).
Currently, novelty feels a bit diffuse.
	5.	Clarify what statistical guarantees the normalization procedure offers.
The exposition in the appendix is correct but difficult to follow. A cleaner explanation would improve readability.

---

> ### Author Response · Authors · 2026-01-31
> **We sincerely thank you for your suggestions and comments**
>
> **Clarify and justify the choice of NLL ranges Z. The paper admits these ranges are subjective. More principled selection criteria (or sensitivity analysis) would strengthen the claims.**
>
> Please refer to the general comments.
>
> **Reduce reliance on small human-annotation segments. Either expand the annotation experiment or avoid using it to argue model superiority.**
>
> We appreciate the reviewer's suggestion. We will relocate the human-annotation experiment to the appendix.
>
> **Provide more discussion of scalability. Since sampling dominates cost, readers need concrete runtime estimates, GPU usage, and guidance for practitioners.**
>
> Please refer to the general comments.
>
> **Position the method more clearly relative to existing alternatives (e.g., LMDiff, dataset-free probing, representation alignment metrics). Currently, novelty feels a bit diffuse.**
>
> LMDiff performs pairwise comparison of model predictions at the token level; however, it remains dataset-based, meaning its analysis is restricted to inputs provided in a fixed corpus. Model-diff complements LMDiff by operating at a higher level: it identifies "interesting candidates" across the model's learned distribution and, crucially, estimates the relative quantity of such samples in the low-NLL region after normalization—something LMDiff cannot provide. To illustrate, consider a teacher-student distillation scenario: a well-trained teacher model and a student model that incorrectly assigns high probability to repetitive-token sequences. Model-diff allows us to quantify the relative proportion of these repetitive-token samples compared to the correctly predicted samples (those around D=0). The training objective is to progressively reduce the number of samples with large |D| values, indicating improved alignment between teacher and student.
>
> Observing perfect D=0 concentration when sampling only from the teacher model's distribution does not guarantee the student model is working well. We propose a critical cross-validation approach where we also sample from the student model's distribution. The key challenge is that we are reasoning about distributional statistics over the input space, not merely inspecting individual samples. If D does not concentrate around 0 when sampling from the student, this reveals that the student model is still making systematic errors. As an analogy: testing a student only on apple samples may yield perfect performance, yet the student might still misclassify oranges as apples—a failure only revealed by testing on the student's own proposed samples. This bidirectional cross-validation critically requires normalization, and our contribution on normalization makes this systematic cross-check possible.
>
> To summarize our positioning relative to existing methods: (1) LMDiff operates at the token level on fixed datasets—Model-diff provides the macro-level distributional view that identifies which regions of the input space warrant token-level analysis; (2) dataset-based evaluation metrics (perplexity, benchmarks) assess performance on human-curated inputs—Model-diff reveals model behaviors in regions humans would not anticipate; (3) the bidirectional cross-validation enabled by our normalization is a unique capability that no existing method provides, as it requires reasoning about distributional statistics rather than individual samples. We will incorporate this positioning more explicitly in the revised manuscript.
>
> We would appreciate it if the reviewer could point us to concrete examples of dataset-free probing and representation alignment metrics, so that we can provide a more targeted comparison in the revision.
>
> **Clarify what statistical guarantees the normalization procedure offers. The exposition in the appendix is correct but difficult to follow. A cleaner explanation would improve readability.**
>
> We appreciate the feedback on readability. We will revise Appendix A.2 to provide a clearer exposition of the statistical guarantees offered by our normalization procedure, including an explicit convergence statement and intuitive explanation of why normalization by the shared intersection |X_{A∩B}| ensures cross-model comparability.
>
> Specifically, the key statistical guarantee is the following: as sample sizes increase, the normalized sampling ratios converge in probability to the ground-truth ratios. This is established by the convergence of |X̃_{A→B}|/|X̃_A| to |X_{A∩B}|/|X_A| (and analogously for model B), which follows from the law of large numbers under PTHR sampling. The shared denominator |X_{A∩B}| ensures both normalized distributions are on the same scale, making their ratio directly comparable to the ground truth. We will incorporate a more mathematically rigorous version of this argument (including a formal theorem and corollary) into the revised Appendix A.2.

---

### Author Response · Authors · 2026-01-31
**General comments**

We sincerely thank you for your suggestions and comments. We have added general comments below to address the common questions raised across reviews.

**Computation resources in details: “Include a section or quantitative data (e.g., wall-clock time, number of sampling steps, convergence plots) analyzing the computational cost and sampling efficiency when applying Model-diff to the models (GPT-2-medium/100-token, LLaMA)” and “Provide more discussion of scalability. Since sampling dominates cost, readers need concrete runtime estimates, GPU usage, and guidance for practitioners.”.**

Our approach is fully independent to existing text generation samplers and requires the same sampling process in the first stage as standard text generation, which has been extensively studied and analyzed in the community (e.g., Section 5.2 "Convergence Analysis for DULA" in https://arxiv.org/pdf/2206.09914).

Following the MCMC stage, we have documented the computational requirements for our Model-diff stage in Appendix C: "We then sample D through the representative inputs within Z with 100,000 steps for GPT-2 experiments and 50,000 steps for other experiments." This will take around 1-2 hours.

Regarding wall-clock time, the MCMC sampling stage has comparable computational cost to standard text generation since it uses the same underlying sampling mechanisms. For reference, llama1 model with 25 sequence length with batch size 64 will take 40 mins per 10000 MC steps on 1 A100 GPU. The Model-diff stage adds minimal overhead as it operates on the sampled representative inputs. We will add explicit timing in the revised manuscript to provide quantitative evidence of computational efficiency.

**Output range: “Clarify and justify the choice of NLL ranges Z. The paper admits these ranges are subjective. More principled selection criteria (or sensitivity analysis) would strengthen the claims.” and “Provide a more rigorous and reproducible method for defining the output range.”**

The definition of the output range is inherently task-dependent and must be tailored to the specific application. To provide a more rigorous and reproducible methodology, we propose the following framework: (1) Define task-specific criteria for what constitutes a valid output (e.g., mathematical expressions, grammatically correct text); (2) Specify the desired confidence level (e.g. 20%)(3) Empirically determine the output range by computing the minimum region that captures the specified percentage of valid samples (e.g. output range containing at least 20% math sequences). This systematic approach allows researchers to adapt the method to different domains while maintaining reproducibility through clearly documented criteria and thresholds. ”

---

### Decision · Action_Editor_yLSh · 2026-03-23

**Recommendation:** Reject

**Audience:**

Yes

**Audience Explanation:**

The problem of comparing language models is well-motivated as language model evaluation is poorly understood as a scientific field.

**Claims And Evidence:**

No

**Claims Explanation:**

Two main concerns remained with reviewers at the end of the reviewing period: (1) the theoretical description of model-diff; and (2) the empirical evidence. I reproduce the concerns of the reviewers here as they will likely be useful for the authors in improving the paper:

> The imprecision of the mathematical description of model-diff makes understanding what really happens impossible. The main imprecisions arise in the discussion of the definition of the output distributions (Section 2.1) and their normalisation (Section 3.3). The final goal initially seems to be approximating the output distributions of the two models. The notion of distribution here is crucial, as the authors mention in the paper and in our discussion that having two distributions allows for an "apples-to-apples" comparison. However, at the start of Section 3.3., the authors change the goal to only being interested in "comparing the inputs whose NLL are low". This change introduces a problem; it is clear how to normalise the output distributions in Equations 1 and 2 to make them comparable, but that is not the case when restricting these functions to the "input values with low NLL".This problem seems to also confuse the authors, because they introduce an additional sum in Equations 5 and 6 with unknown summation domain. When asked in the discussion about the domain of this sum, they mention it is another sum over some some set of, yet does not even depend on any more. The remaining exposition, as well as our discussion, is then based on vague intuitions of "obtaining representations with a similar scale" that make any understanding of what is actually computed impossible.

> The empirical toy validation is essential and at the same time not representative of a practical application of model-diff. The toy experiment has been clarified by the authors during our discussion, which convinced me about its necessity. However, the setup of the experiment does not allow to draw the conclusions the authors want to draw. If the goal is to check the approximation quality of Model-diff by considering a case where the entire input space is enumerable, then something has to at least be approximated. Instead, the authors opt to draw as many samples as there are input options, which is almost identical to simply enumerating. Importantly, other experiments only allow to sample a fraction of a more practical input space. To truly empirically show that Model-diff soundly approximates the true quantity in practice, a more complete study with a varying number of samples, including realistic ratios, is necessary. I would be able to disregard the need for empirical evidence that Model-diff uses sound approximations, but the theory sadly does not allow that.

>Additionally, the connection to recall is, as also mentioned by Reviewer 7SrL, overly strong and out-of-place. Other minor claims in the experimental discussion are speculative and have no solid empirical or theoretical foundation, e.g. the plausability of plagiarism checks.

I agree with the concerns raised around clarity of the theoretical results and I do not see a way to address these within this review cycle. I additionally agree with the concerns around the empirical results: the authors have not demonstrated that this approach scales to real-world language models of interest and it is not obvious how they can easily change the experiments to do this. For these reasons, I vote to reject.